# LncRNA-LncDACH1 mediated phenotypic switching of smooth muscle cells during neointimal hyperplasia in male arteriovenous fistulas

Zhaozheng Li [1,5], Yao Zhao[1,5], Zhenwei Pan [2,3], Benzhi Cai [2,3,4], Chengwei Zhang [1,6] ✉ & Jundong Jiao[1,6] ✉

Arteriovenous fistulas (AVFs) are the most common vascular access points for hemodialysis (HD), but they have a high incidence of postoperative dysfunction, mainly due to excessive neointimal hyperplasia (NIH). Our previous studies have revealed a highly conserved LncRNA-LncDACH1 as an important regulator of cardiomyocyte and fibroblast proliferation. Herein, we find that LncDACH1 regulates NIH in AVF in male mice with conditional knockout of smooth muscle cell-specific LncDACH1 and in male mice model of AVF with LncDACH1 overexpression by adeno-associated virus. Mechanistically, silence of LncDACH1 activates p-AKT through promoting the expression of heat shock protein 90 (HSP90) and serine/arginine-rich splicing factor protein kinase 1 (SRPK1). Moreover, LncDACH1 is transcriptionally activated by transcription factor KLF9 that binds directly to the promoter region of the LncDACH1 gene. In this work, during AVF NIH, LncDACH1 is downregulated by KLF9 and promotes NIH through the HSP90/ SRPK1/ AKT signaling axis.

End-stage renal disease (ESRD) is a serious public health concern and the number of patients worldwide is increasing every year[1]. Patients with ESRD primarily receive hemodialysis (HD) as renal replacement therapy. Arteriovenous fistulas (AVFs) are the preferred vascular access points for HD. AVFs are associated with lower rates of postoperative infections and fewer complications than other techniques such as arteriovenous grafts[2,3]. The success of HD depends on proper AVF function; however, nearly 50% of AVFs fail to provide effective HD two years after their establishment due to the occurrence of postoperative dysfunction[4].

Neointimal hyperplasia (NIH) is a major cause of AVF postoperative dysfunction. Previous studies have shown that the abnormal proliferation and migration of vascular smooth muscle cells (VSMCs), which leads to NIH, is a hallmark of AVF maturation failure[5,6]. The aberrant proliferation and migration of VSMCs are closely linked to their phenotypic switching capacities. Specifically, VSMCs can convert from a differentiated phenotype to a dedifferentiated phenotype in response to stimulation by growth factors such as platelet-derived growth factor-BB (PDGF-BB) and tumor necrosis factor-alpha (TNF-α). The differentiated phenotype is characterized by low proliferative and migratory capacities and high expression of differentiation markers such as α-smooth muscle actin (α-SMA) and smooth muscle 22α (SM22α). In contrast, the dedifferentiated phenotype is characterized by high proliferative and migratory capacities and high expression of dedifferentiation markers such as osteopontin (Opn) and vimentin[7,8]. The mechanisms

[1]Department of Nephrology, The Second Affiliated Hospital of Harbin Medical University, 150086 Harbin, China. [2]Department of Pharmacy at The Second Affiliated Hospital, Harbin Medical University, 150086 Harbin, China. [3]Department of Pharmacology (The Key Laboratory of Cardiovascular Medicine Research, Ministry of Education) at College of Pharmacy, Harbin Medical University, 150086 Harbin, China. [4]Department of Clinical Pharmacology (the Heilongjiang Key Laboratory of Drug Research), Harbin Medical University, 150086 Harbin, China. [5]These authors contributed equally: Zhaozheng Li, Yao Zhao. [6]These authors jointly supervised this work: Chengwei Zhang, Jundong Jiao. ✉e-mail: zhangchengwei1981@163.com; jiaojundong@163.com

underlying VSMC phenotypic switching are not yet fully understood. Therefore, a better understanding of the molecular mechanisms leading to VSMC phenotypic switching during NIH is essential for designing therapeutic strategies.

Long noncoding RNAs (LncRNAs) are a newly discovered class of RNAs >200 nucleotides long that lack protein-coding properties[9]. LncRNAs regulate a variety of biological processes and are involved in the pathogenesis of various diseases, including cardiovascular disease[10]. Recently, our group have reported that a novel LncRNA, LncDACH1, is important for cardiac repair and regeneration after heart failure and myocardial infarction[11,12]. We further also reported that LncDACH1 promotes the development of idiopathic pulmonary fibrosis by regulating the proliferation and migration of lung fibroblasts[13]. However, the mechanisms underlying the role and regulation of LncDACH1 in vascular diseases are still unknown, particularly in the context of vascular NIH.

In the present study, we observed that the highly conserved LncRNA LncDACH1 was downregulated during NIH and VSMC dedifferentiation. Further studies revealed that conditional knockout (CKO) LncDACH1 mice experienced exacerbated AVF NIH, whereas adeno-associated virus-LncDACH1 (AAV-LncDACH1) overexpressing mice exhibited attenuated AVF NIH. Mechanistically, LncDACH1 inhibited serine/arginine-rich splicing factor protein kinase 1 (SRPK1) and heat shock protein 90 (HSP90) to regulate p-AKT. Using RNA Immunoprecipitation (RIP) assay, we found that LncDACH1 directly bound SRPK1 but not HSP90. Our data also uncovered that LncDACH1 regulated nuclear translocation of SRPK1 by targeting HSP90. These results prompted us to posit that LncDACH1 controls the nuclear translocation of SRPK1 by regulating the binding and interactions between HSP90 and SRPK1, which may be a key mechanism underlying the cellular function of LncDACH1. Additionally, the transcription factor KLF9 acted as a transactivator to positively regulate LncDACH1 transcription by binding directly to the promoter region of the LncDACH1 gene.

## Results

### LncDACH1 expression is downregulated during NIH and VSMC phenotypic switching

First, to explore if LncDACH1 is involved in AVF, we collected the samples from human and mouse AVF tissues. We found that the stenosis veins of CKD patients with NIH were more hyperplastic than the preoperative veins by HE staining (Fig. 1a) and morphometric analysis (Fig. 1b; Supplementary Fig. 1a, b). Subsequently, to characterize changes in LncDACH1 expression, we performed qRT-PCR on preoperative and stenosis vein samples from CKD patients with NIH. We found that LncDACH1 expression was downregulated in stenosis veins compared to preoperative veins (Fig. 1c).

Next, we established an AVF mouse model based on methods described in previous studies and validated it by H&E staining (Fig. 1d) and morphometric analysis (Fig. 1e; Supplementary Fig. 1c–f)[14–16]. We then performed qRT-PCR and find that LncDACH1 expression was downregulated in a time-dependent manner after the establishment of AVF (Fig. 1f). We therefore hypothesized that LncDACH1 may be involved in the pathogenesis of NIH. Subsequently, we performed qRT-PCR and found that LncDACH1 was expressed in a variety of mouse tissues (Fig. 1g). We also examined the expression of LncDACH1 in VSMCs by fluorescence in situ hybridization (Fig. 1h) and qRT-PCR (Fig. 1I) and found that it was distributed in both the cytoplasm and nucleus.

Finally, we used PDGF-BB to induce dedifferentiation in human (Supplementary Fig. 1g) and mouse VSMCs. With increasing PDGF-BB concentrations (0, 5, 10, and 20 ng/mL), LncDACH1 expression in VSMCs decreased in a dose-dependent manner (Fig. 1j, l). In addition, VSMC LncDACH1 expression exhibited time-dependent reductions when PDGF-BB (10 ng/mL) was administered for different periods of time (24, 48, and 72 h) (Fig. 1k, m). These data suggest a relationship between LncDACH1 expression and PDGF-BB-induced VSMC dedifferentiation. Since LncDACH1 was more significantly downregulated in mouse VSMCs during dedifferentiation, cells from this species were used for subsequent in vitro experiments.

### Overexpression of LncDACH1 inhibits proliferation, migration, and phenotypic switching during VSMC dedifferentiation

We next performed functional experiments to assess the potential role of LncDACH1 in VSMC proliferation, migration, and phenotypic switching. We established and validated a plasmid that stably overexpressed LncDACH1. Forty-eight hours after dedifferentiated VSMCs were transfected with this plasmid, the expression of LncDACH1 was significantly increased compared to cells that received an empty vector plasmid (Fig. 2a). We found that overexpression of LncDACH1 inhibited VSMC proliferation by the cell counting kit-8 (CCK-8) (Fig. 2b) and EdU assays (Fig. 2c, d). Moreover, wound-healing (Fig. 2e) and transwell assays (Fig. 2f) showed that cell migration was also significantly inhibited by LncDACH1 overexpression. Given that phenotypic switching is closely associated with VSMC proliferation and migration, we further examined the role of LncDACH1 upregulation in this process. First, we determined the successful dedifferentiation of VMSCs after stimulation with PDGF-BB by validating differentiation phenotypic markers α-SMA and SM22α and de-differentiation phenotypic markers Opn and vimentin. Next, the protein levels of the differentiation markers were upregulated following LncDACH1 overexpression, whereas the protein levels of the dedifferentiation markers were downregulated (Fig. 2g–k). In conclusion, these results suggest that overexpression of LncDACH1 inhibits proliferation, migration, and phenotypic switching during VSMC dedifferentiation.

### Silencing LncDACH1 promotes VSMC proliferation, migration, and phenotypic switching during dedifferentiation

To further investigate the function of LncDACH1 in VSMCs, we designed and synthesized small interfering RNAs (siRNAs) to reduce LncDACH1 expression. The expression of LncDACH1 was significantly downregulated in dedifferentiated VSMCs after transfection with si-LncDACH1 compared to cells that received an si-NC (Fig. 3a). We found that LncDACH1 silencing promoted VSMC proliferation by the CCK-8 (Fig. 3b) and EdU (Fig. 3c, d) assays. Additionally, wound-healing (Fig. 3e) and transwell (Fig. 3f) assays showed that LncDACH1 silencing enhanced the migratory capacity of VSMCs. Consistent with these results, the protein levels of differentiation markers were all downregulated, whereas the protein levels of dedifferentiation markers were upregulated (Fig. 3g–k). These results suggest that LncDACH1-silencing promotes proliferation, migration, and phenotypic switching during VSMC dedifferentiation.

### Modulation of LncDACH1 in differentiated VSMCs does not affect their proliferation, migration

To determine whether LncDACH1 has a regulatory role in differentiated VSMCs, we overexpressed or silenced LncDACH1 in these cells. Compared to their respective controls, we found that overexpression or silencing of LncDACH1 did not affect the proliferation (Supplementary Fig. 2a–c) or migration (Supplementary Fig. 2d–g) of differentiated VSMCs. These results suggest that LncDACH1 selectively regulates the proliferation and migration of dedifferentiated VSMCs.

### LncDACH1 promotes dedifferentiated VSMC proliferation, migration, and phenotypic switching by upregulating HSP90

To explore the molecular mechanisms by which LncDACH1 modulates VSMC proliferation, migration, and dedifferentiation, we performed iTRAQ quantitative protein profiling and bioinformatic analyses to

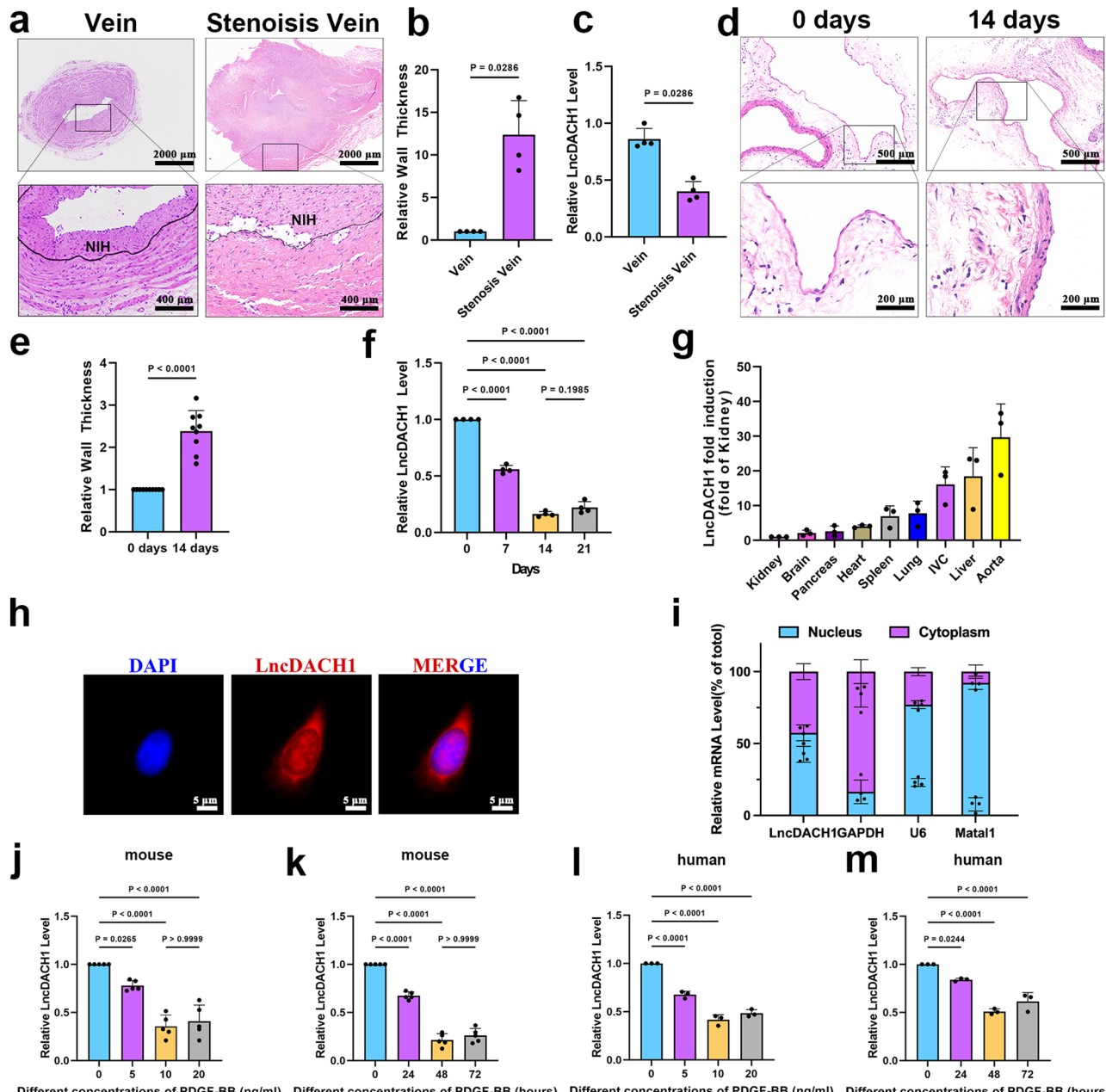

**Fig. 1 | LncDACH1 expression is downregulated during NIH and VSMC phenotypic switching. a–c** HE staining, morphometric analysis and qRT-PCR for LncDACH1 expression of preoperative veins (Scale bar, 2000 μm) in human AVF versus stenosis veins (Scale bar, 400 μm) after AVF surgery (n = 4 at each group). **d, e** Pre-AVF vein (n = 10) and 14-day post-AVF vein (n = 9) HE staining and morphometric analysis in mouse. Scale bar, 200 μm. **f** Venous tissues were obtained from the AVF mouse model at 0, 1, 2, and 3 weeks (n = 4 at each time point) and qRT-PCR was applied to detect the expression levels of LncDACH1 during NIH in the AVF mouse model. **g** The expression abundance of LncDACH1 in mouse tissues was measured by qRT- PCR (n = 4). **h** FISH was used to detect the distribution of LncDACH1 in VSMC. Scale bar, 5 μm. **i** mRNA levels of LncDACH1, GAPDH, U6, and Matal1 in VSMC cytoplasm and nuclear fractions were measured by qRT-PCR, respectively. **j, l** LncDACH1 expression levels in mouse VSMC (n = 5) and humans VSMC (n = 3) were measured by qRT-PCR after stimulation with different concentration gradients (0, 5, 10, and 20 ng/mL) of PDGF-BB for 48 h. **k, m** LncDACH1 expression levels in mouse VSMC (n = 5) and humans VSMC (n = 3) were measured by qRT-PCR after stimulation with the same concentration of PDGF-BB (10 ng/mL) at different times (24, 48, and 72 h) (n = 5). If not otherwise specified, VSMC induction using PDGF-BB in this study were induced with 10 ng/mL for 48 h. PDGF-BB (P), neointimal hyperplasia (NIH). The n numbers represent biologically independent samples. Data are presented as mean values ± SD (**b, c, e, f, j–m**). *P*-values were determined by two-sided nonparametric tests (**b, c, e**) and two-sided one-way ANOVA (**f, j–i**, m) by Bonferroni's multiple comparisons test. Source data are provided as a Source Data file.

identify downstream proteins affected by LncDACH1 (Supplementary Fig. 3a–d). Among them, the total HSP90 protein expression levels in the enriched PI3K/AKT signaling pathway were regulated by LncDACH1 (Supplementary Fig. 3e–h). We therefore overexpressed LncDACH1 in dedifferentiated VSMCs and found that the total protein levels of HSP90 were downregulated (Fig. 4a); in contrast, HSP90 protein levels were upregulated after LncDACH1 silencing (Fig. 4b).

Subsequently, we silenced HSP90 in VSMCs and verified the silencing efficiency (Supplementary Fig. 4a). We then divided VSMCs into the following groups: si-NC, si-LncDACH1, si-HSP90, and si-LncDACH1 + si-HSP90. We found that silencing HSP90 reduced p-AKT expression (Fig. 4c), whereas simultaneous silencing of LncDACH1 and HSP90 partially reversed the VSMC proliferation (Fig. 4d–f), migration (Fig. 4g–i), and dedifferentiation (Fig. 4j, k) induced by LncDACH1

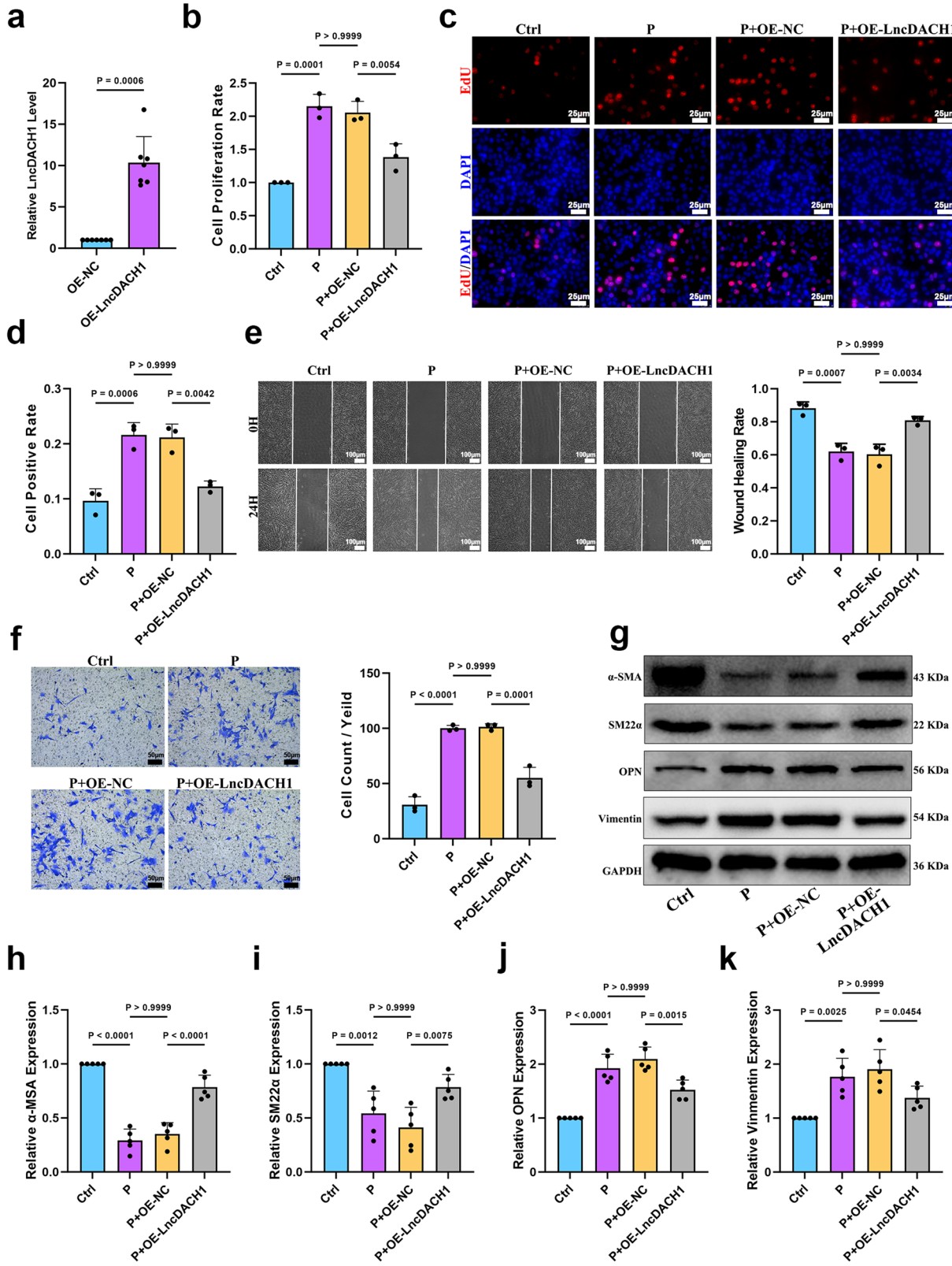

silencing. These results show that silencing of LncDACH1 promotes the proliferation, migration, and phenotypic switching of VSMCs during dedifferentiation by upregulating HSP90 and activating p-AKT. Finally, RNA immunoprecipitation (RIP) revealed that LncDACH1-mediated regulation of HSP90 protein levels was achieved through an indirect mechanism rather than by direct binding (Fig. 4l).

## LncDACH1 binds to SRPK1 protein

To further explore the molecular mechanism by which LncDACH1 regulates VSMC proliferation, migration, and phenotypic switching, we used an RNA pull-down assay and LC-MS in VSMCs to screen for proteins that may bind to LncDACH1, including SRPK1. VSMC lysates were incubated with biotinylated LncDACH1 or antisense RNA

**Fig. 2 | Overexpression of LncDACH1 inhibits proliferation, migration, and phenotypic switching during VSMC dedifferentiation. a** The transfection efficiency of pcDNA3.0 LncDACH1 in VSMC was measured by qRT-PCR ($n = 7$). **b** The effect of overexpression of LncDACH1 on the proliferation capacity of VSMC was examined by CCK-8 assay ($n = 3$). **c, d** Effect of overexpression of LncDACH1 on the proliferation capacity of VSMC was examined by EdU staining assay ($n = 3$). Scale bar, 25 μm. **e** Effect on VSMC migration capacity after overexpression of LncDACH1 was examined by Wound Healing assay ($n = 3$). Scale bar, 100 μm. **f** Effect on VSMC migration capacity after overexpression of LncDACH1 was examined by Transwell assay ($n = 3$). Scale bar, 50 μm. **g–k** Effect of overexpression of LncDACH1 on VSMC differentiation phenotype markers and dedifferentiation phenotype marker protein levels by Western Blot assay ($n = 5$). PDGF-BB (P), control (CTRL), negative control (NC), overexpression (OE). The $n$ numbers represent biologically independent samples. Data are presented as mean values ± SD (**a, b, d, e, f, h–k**). *P*-values were determined by two-sided nonparametric tests (**a**) and two-sided one-way ANOVA (**b, d–f** and **h–k**) by Bonferroni's multiple comparisons test. Source data are provided as a Source Data file.

probes for RNA pull down, and a western blot with SRPK1 antibodies was performed (Fig. 5a). Subsequently, we used RIP to validate the interaction between LncDACH1 and SRPK1 (Fig. 5b). This result was supported by predictions from the RPISeq database[http://pridb.gdcb.iastate.edu/RPISeq/] (Supplementary Fig. 5a)[17].

To characterize the molecular basis of the LncDACH1-SRPK1 interaction, we sought to identify the specific binding fragment on LncDACH1. We therefore constructed different LncDACH1 fragments (LncDACH1-A–E) (Fig. 5c). RNA pull-down experiments revealed that four LncDACH1 fragments, A, B, C, and E, interacted with SRPK1, whereas LncDACH1-D did not (Fig. 5d). This result suggests that LncDACH1-E (nucleotides 774–1251) contains the SRPK1 binding region. catRAPID[https://tartaglialab.com/page/catrapid_group], an algorithm that predicts interactions between peptide and nucleotide sequence fragments, also predicted that LncDACH1-E interacts with SRPK1. Specifically, catRAPID fragmentation analysis revealed that nucleotide positions 814–897 of LncDACH1 have a high likelihood of binding to amino acid residues 101–152 of SRPK1 (Supplementary Fig. 5b). Therefore, we investigated whether mutation of the LncDACH1 binding site (LncDACH1-F; 774–1251 mutant [MUT]) would reduce the direct binding capacity of LncDACH1 to SRPK1. RNA pull-down experiments showed that LncDACH1-F was unable to interact with SRPK1 (Fig. 5e). Taken together, these data suggest that LncDACH1 binds directly to SRPK1 via nucleotides 774–1251(Supplementary Fig. 5c).

**LncDACH1 promotes dedifferentiated VSMC proliferation, migration, and phenotypic switching by upregulating SRPK1**

To explore the molecular mechanism, we overexpressed LncDACH1 in dedifferentiated VSMCs and found that the total protein levels of SRPK1 were reduced (Fig. 5f). In contrast, the levels of SRPK1 were increased after LncDACH1 silencing (Fig. 5g). Subsequently, we verified the silencing efficiency of si-SRPK1 in VSMCs (Supplementary Fig. 6a). We then divided VSMCs into the following groups: si-NC, si-LncDACH1, si-SRPK1, and si-LncDACH1 + si-SRPK1. We found that SRPK1 silencing inhibited p-AKT expression (Fig. 5h), whereas the simultaneous downregulation of LncDACH1 and SRPK1 partially reversed the proliferation, (Fig. 5i–k), migration (Fig. 5l, m), and dedifferentiation (Fig. 5n) of VSMCs induced by LncDACH1 suppression. These results show that silencing of LncDACH1 promotes the proliferation, migration, and phenotypic switching of dedifferentiated VSMCs by upregulating SRPK1 and activating p-AKT.

**HSP90 mediates LncDACH1-induced nuclear translocation of SRPK1**

We next determined whether LncDACH1 regulates SRPK1 nuclear translocation by modulating HSP90. First, a cellular immunofluorescence assay revealed that LncDACH1 silencing significantly inhibited SRPK1 nuclear translocation, whereas silencing HSP90 alleviated the inhibitory effect of LncDACH1 on SRPK1 nuclear translocation (Fig. 6a). This observation was confirmed by western blot analysis of isolated cytoplasmic and nuclear proteins (Fig. 6b). The VSMCs were divided into the si-NC group and the si-LncDACH1 group. Through Co-IP experiments, we found a weak mutual binding ability between HSP90 and SRPK1 in the si-NC group, while silencing

LncDACH1 could enhance the binding ability between the two (Fig. 6c). This suggests that LncDACH1 may regulate SRPK1 nuclear translocation by modulating the binding ability between HSP90 and SRPK1.

**KLF9 positively regulates LncDACH1 transcription**

To understand why LncDACH1 expression is downregulated in dedifferentiated VSMCs, we performed a computational analysis with the UCSC Genome Browser[https://genome.ucsc.edu][18] and JASPAR prediction tools[https://jaspar.genereg.net] (Supplementary Fig. 7a)[19]. Based on our results, we hypothesized that KLF9 directly interacts with the LncDACH1 promoter (Fig. 7a). First, we cloned the full-length (FL) LncDACH1 promoter (2 kb) into a luciferase reporter plasmid. The 2kbFL reporter gene was then co-transfected with KLF9 overexpression plasmid or overexpression plasmid empty vector into HEK-293T cells. Analysis of the resulting relative luminescence revealed positive response when the 2kbFL reporter gene was co-transfected with the KLF9 overexpression plasmid (Fig. 7b). We then truncated the 2kbFL reporter gene into three fragments: P1 (nucleotides 684–2000), P2 (nucleotides 1099–2000), and P3 (nucleotides 1500–2000). These fragments were selected based on the predicted KLF9 binding site in the LncDACH1 promoter identified through JASPAR analysis. These fragments were then co-transfected with the KLF9 overexpression plasmid or the empty vector control. In this assay, P1 showed positive response while P2 and P3 showed negative response (Fig. 7c). These results suggest that KLF9 binds to the LncDACH1 promoter between nucleotides 684 and 1099; JASPAR predicted a binding site between nucleotides 888 and 928. To verify this result, we constructed a mutant 2kbFL vector (MUT) in which nucleotides 888-928 were mutated. The wild type 2kbFL or MUT 2kbFL plasmid was then co-transfected with the KLF9 overexpression plasmid or empty vector control. In this analysis, positive response was obtained for wild type 2kbFL and negative response were obtained for MUT 2kbFL plasmid (Fig. 7d). These findings suggest that KLF9 binds to the LncDACH1 promoter between nucleotides 888 and 928.

Subsequently, we performed ChIP-qPCR to detect the binding of KLF9 to nucleotides 888–928 of the LncDACH1 promoter. We designed two qPCR probe sets: a negative control (NC) probe located 5927–5996 nucleotides downstream of the LncDACH1 transcription start site (TSS) and a test probe (pmLncDACH1) located 858–942 nucleotides upstream of the LncDACH1 TSS (Fig. 7e). We then transfected a KLF9-Flag plasmid into VSMCs and pulled down chromatin with anti-Flag antibodies. qPCR analysis of pulled-down chromatin showed that KLF9 was bound to the LncDACH1 promoter between nucleotides 858 and 942 (Fig. 7f, g).

Finally, we found that KLF9 mRNA expression levels were downregulated during VSMC dedifferentiation, consistent with the LncDACH1 trend (Fig. 7h). We then designed KLF9 siRNAs and transfected them into dedifferentiated VSMCs (Supplementary Fig. 8a), which downregulated LncDACH1 expression (Fig. 7i). The KLF9 overexpression plasmid was then transfected into these cells (Supplementary Fig. 8b), which resulted in LncDACH1 upregulation (Fig. 7j). Taken together, these results confirm our hypothesis that KLF9 positively regulates LncDACH1 transcription.

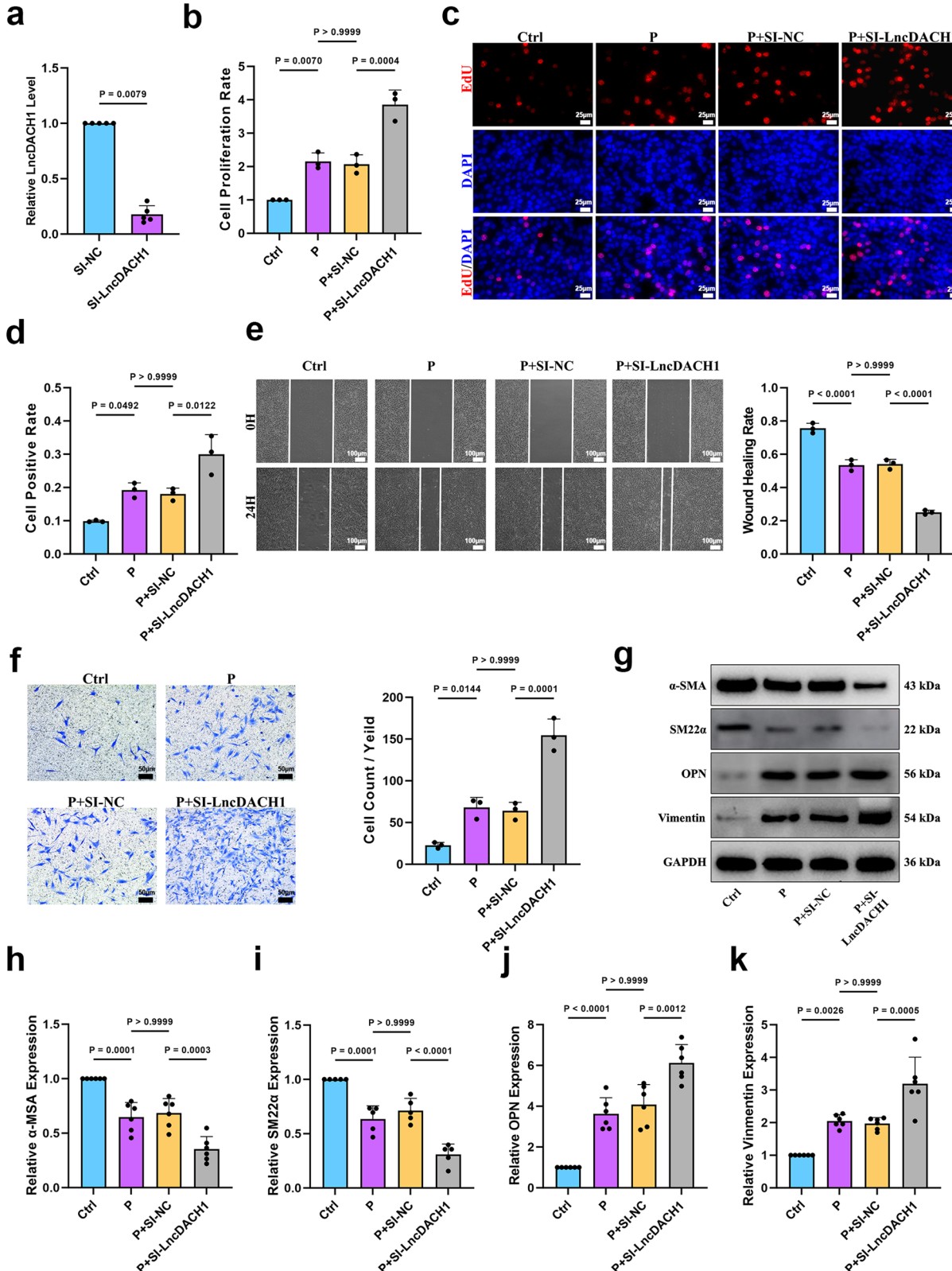

## SMC-specific LncDACH1 conditional knockout (CKO) mice exhibit aggravating NIH in AVF

We generated LncDACH1 CKO mice using the LoxP/Cre strategy to verify the effect of LncDACH1 on NIH (Fig. 8a). LncDACH1 CKO mice were generated by crossing LncDACH1fl/fl mice with SMMHC-CreERT2 mice. Using qRT-PCR, we verified that significantly lower levels of LncDACH1 were observed in the vessels of LncDACH1 CKO compared

to negative control mice (Supplementary Fig. 9a). We also used H&E staining and morphometric analysis to compare the veins of mice with and without established AVF (Supplementary Fig. 10a–h). No significant differences in the thicknesses of NIH were found in the veins of LncDACH1 CKO mouse mice without established AVF compared to their respective negative control mice (Fig. 8c). However, NIH thickness was increased in the veins of LncDACH1 CKO mice with

**Fig. 3 | Silencing LncDACH1 promotes VSMC proliferation, migration, and phenotypic switching during dedifferentiation. a** Transfection efficiency of si-LncDACH1 in VSMC was examined using qRT-PCR ($n = 5$). **b** Effect of silencing LncDACH1 on the proliferation capacity of VSMC was examined by CCK-8 assay ($n = 3$). **c, d** Effect of silencing LncDACH1 on the proliferation capacity of VSMC was examined by EdU staining assay ($n = 3$). Scale bar, 25 μm. **e** Effect on VSMC migration capacity after silencing LncDACH1 was tested by Wound Healing assay. Scale bar, 100 μm ($n = 3$). **f** Effect on VSMC migration capacity after silencing LncDACH1 was examined by Transwell assay ($n = 3$). Scale bar, 50 μm. **g–k** Effect of silencing LncDACH1 on VSMC differentiation phenotype markers and dedifferentiation phenotype marker protein levels by Western Blot assay ($n = 5$). PDGF-BB (P), control (CTRL), negative control (NC), small interfering (SI). The $n$ numbers represent biologically independent samples. Data are presented as mean values ± SD (**a, b, d–f, h–k**). *P*-values were determined by two-sided nonparametric tests (a) and two-sided one-way ANOVA (**b, d–f** and **h–k**) by Bonferroni's multiple comparisons test. Source data are provided as a Source Data file.

established AVF when compared to the negative control mice with established AVF (Fig. 8e). Using western blot analysis of total venous tissue protein lysates from mice with AVF, we observed upregulation of SRPK1 and HSP90 levels in LncDACH1 CKO mice compared to negative control mice (Fig. 8f). These results suggest that CKO of LncDACH1 exacerbates NIH in AVF mice.

### Overexpression of LncDACH1 in vivo inhibits NIH in the context of AVF

We topically applied a mixture of biogel and AAV-LncDACH1 to the vascular surfaces of mice to verify the effect of LncDACH1 overexpression on NIH (Fig. 8b). The upregulation of LncDACH1 expression in the vessels of AAV-LncDACH1 mice compared to AAV-NC mice was verified by qRT-PCR (Supplementary Fig. 9b). We then compared the veins of mice without established AVF to those of animals with established AVF using H&E staining and morphometric analyses (Supplementary Fig. 10i–p). No significant differences were found in the thicknesses of venous NIH in AAV-LncDACH1 mice without established AVF compared to their respective negative controls (Fig. 8d). However, venous NIH thickness was reduced in AAV-LncDACH1 mice with established AVF when compared to the negative control mice with established AVF (Fig. 8g). Using western blot analysis of total venous tissue protein lysates from mice with AVF, we observed downregulation of SRPK1 and HSP90 levels in AAV-LncDACH1 mice compared to AAV-NC mice (Fig. 8h). These results suggest that overexpression of LncDACH1 can reduce the extent of NIH in AVF mice.

### LncDACH1 affected NIH in the context of AVF by regulating VSMC phenotype switching

To further clarify the regulatory mechanism of LncDACH1 in the NIH process of AVF mice, first, we examined the cell proliferation-related marker Ki67 and apoptosis-related marker Tunel by immunohistochemistry and immunofluorescence in the samples of 0 days before AVF and 14 days after AVF, respectively (Fig. 9a, f). As we expected, the expression level of Ki67 was increased, and that of Tunel was decreased at 14 days compared to 0 days (Supplementary Fig. 11a, f). At the same time, we examined the endothelial cell phenotype marker CD31, fibroblast phenotype marker FSP-1, macrophage phenotype marker CD68 and VSMC phenotype switching markers α-SMA and Vimentin (Fig. 9a). The results showed that no significant difference in the expression level of CD31 was observed at 14 days compared to 0 days, while the expression levels of FSP-1, CD68, α-SMA and Vimentin were increased (Supplementary Fig. 11a). We continued to examine the above markers in samples from AAV-LncDACH1 and CKO-LncDACH1 mice without established AVF (Fig. 9b, c, f). We found that modulation of LncDACH1 did not affect changes in the expression levels of these markers without established AVF, which is also consistent with the fact that LncDACH1 does not affect NIH without established AVF (Supplementary Fig. 11b, c, f).

Subsequently, we also examined the above markers in AAV-LncDACH1 and CKO-LncDACH1 mice with established AVF (Fig. 9d–f). We found that Ki67 expression levels increased and Tunel expression levels decreased after specific knockdown of LncDACH1 compared to controls (Supplementary Fig. 11e, f). The opposite was true after overexpression of LncDACH1 (Supplementary Fig. 11d, f).

This is also consistent with the findings that specific knockdown of LncDACH1 aggravated NIH, and overexpression of LncDACH1 inhibits NIH. For the remaining markers, we found that LncDACH1 did not regulate CD31 expression levels. In contrast, specific knockdown of LncDACH1 increased the expression levels of FSP-1, CD68, α-SMA, and Vimentin (Supplementary Fig. 11e). The opposite after overexpression of LncDACH1 (Supplementary Fig. 11d). Finally, we again verified the effect of modulating LncDACH1 after establishing AVF on the expression levels of phenotype markers by Western Blot assay. We found that specific knockdown of LncDACH1 decreased the expression level of SM22α while increasing the expression level of Vimentin and OPN, in contrast to overexpression of LncDACH1 (Supplementary Fig. 12a, b). These results all confirm that LncDACH1 affected NIH in the context of AVF by regulating VSMC phenotype switching.

## Discussion

In the present study, our primary finding was that LncDACH1 expression was downregulated during VSMC dedifferentiation and NIH in the context of AVF. In vivo experiments also revealed that LncDACH1 CKO mice with AVF exhibited exacerbated NIH, whereas AAV-LncDACH1 mice mouse experienced attenuated AVF NIH. Mechanistically, silencing LncDACH1 promoted p-AKT activation by upregulating HSP90. Additionally, we found that one key fragment of LncDACH1 bound directly to SRPK1 and that LncDACH1 silencing led to SRPK1 upregulation and p-AKT activation. Moreover, LncDACH1 inhibited SRPK1 nuclear translocation by upregulating HSP90. Finally, KLF9 bound directly to the LncDACH1 promoter and positively regulated its transcription. Taken together, these data suggest that LncDACH1 may be a potential therapeutic target for NIH.

Excessive NIH is a pathophysiological process that leads to AVF dysfunction. NIH involves several key cellular processes including inflammatory cell infiltration, myofibroblast activation, overproduction of extracellular matrix, impaired endothelial cell function, and VSMC phenotypic switching[20]. VSMC phenotypic switching is a particularly important cause of excessive NIH[21]. During NIH formation, VSMCs switch from a contractile phenotype to a synthetic phenotype and acquire greater proliferative and migratory capacities[22]. In recent years, there has been growing interest in LncRNA-mediated regulation of VSMC phenotypic switching. For example, Yao et al. observed that LncRNA XR007793 regulates VSMC proliferation and migration and is involved in vascular remodeling during hypertension[23]. Additionally, Ahmed et al. reported that LncRNA NEAT1 promoting VSMC phenotypic switching by binding to WDR5[24]. Therefore, it is important to explore the molecular mechanisms by which LncRNAs regulate VSMC phenotypic switching.

To understand the molecular mechanism by which LncDACH1 regulates VSMC phenotypic switching, we performed iTRAQ quantitative protein profiling on VSMCs transfected with si-LncDACH1 or si-NC. Through bioinformatic analysis of differential protein expression, we found that the PI3K/AKT signaling pathway was highly enriched. AKT family kinases (also known as protein kinase B/PKB) are mammalian serine/threonine kinases with high degrees of structural and functional conservation. AKT phosphorylation levels are regulated by a variety of kinases and phosphatases, and p-AKT is

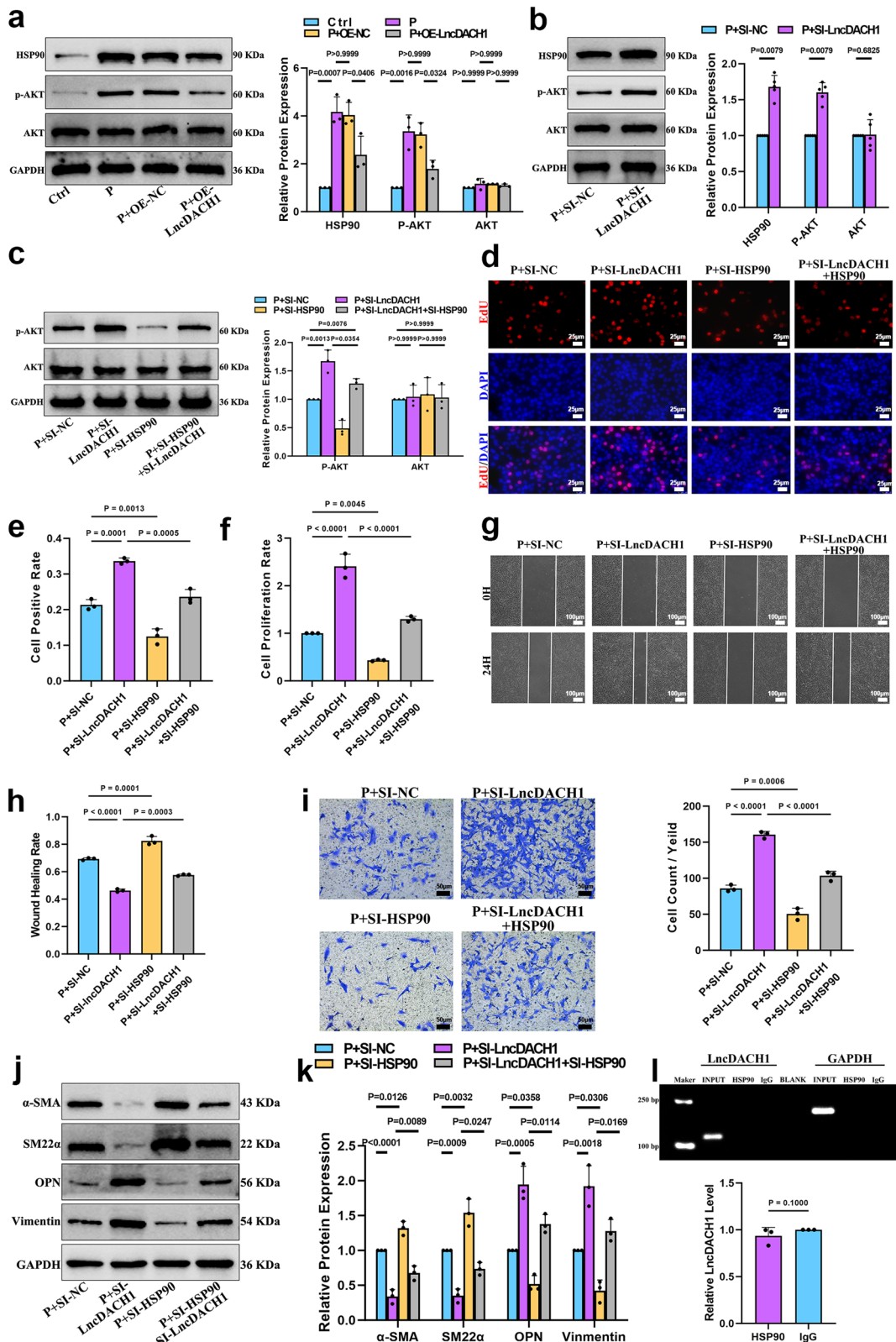

thought to control inflammation and cell migration, proliferation, and apoptosis[25]. Previous work has shown that p-AKT plays a key role in excessive NIH and VSMC proliferation, migration, and phenotypic switching[26]. HSP90 was among the most notable PI3K/AKT-related proteins enriched in si-LncDACH1-transfected cells. HSP90 is a highly conserved molecular chaperone that assembles various client proteins and regulates their folding, assembly, signal transduction,

translocation, and transcription. Importantly, AKT is an established HSP90 client protein[27]. We silenced LncDACH1 in dedifferentiated VSMCs and observed increases in HSP90 levels; conversely, over-expression of LncDACH1 resulted in HSP90 downregulation, consistent with our iTRAQ quantitative protein profiling results. Next, we verified that HSP90 modulates LncDACH1-mediated VSMC proliferation, migration, and phenotypic switching by regulating AKT

**Fig. 4 | LncDACH1 promotes dedifferentiated VSMC proliferation, migration, and phenotypic switching by upregulating HSP90. a, b** Protein expression levels of HSP90, p-AKT and AKT after overexpression (*n* = 3) or silencing (*n* = 5) of LncDACH1 in VSMC were detected using Western Blot. VSMC is divided into the following four groups: si-NC, si-LncDACH1, si-HSP90, si-LncDACH1+si-HSP90. **c** p-AKT and AKT protein expression levels were detected by Western Blot (*n* = 3). **d, e** The proliferation capacity of VSMC was tested by EdU staining assay (*n* = 3). Scale bar, 25 μm. **f** The proliferation capacity of VSMC was examined by CCK8 assay (*n* = 3). **g, h** The ability of VSMC to migrate was tested by Wound Healing assays (*n* = 3). Scale bar, 100 μm. **i** The ability of VSMC to migrate was tested by Transwell assay (*n* = 3). Scale bar, 50 μm. **j, k** Protein expression levels of VSMC differentiation phenotype markers and dedifferentiation phenotype markers were measured by Western Blot assay (*n* = 3). **l** RIP was used to assess the binding capacity between LncDACH1 and HSP90 proteins. PDGF-BB (P), control (CTRL), negative control (NC), overexpression (OE), small interfering (SI). The *n* numbers represent biologically independent samples. Data are presented as mean values ± SD (**a**–**c**, **e**, **f**, **h**, **i**, **k**, **l**). *P*-values were determined by two-sided nonparametric tests (**b**, **l**) and two-sided one-way ANOVA (**a**, **c**, **e**, **f**, **h**, **i**, **k**) by Bonferroni's multiple comparisons test. Source data are provided as a Source Data file.

activity. Previous work has demonstrated that HSP90 regulates AKT activity by modulating the AKT phosphatase PP2A, and that overexpression of HSP90 regulates the pathogenesis of multiple diseases by activating p-AKT[28]. By simultaneously silencing LncDACH1 and HSP90, we were able to partially reverse the promoting effect of LncDACH1 on p-AKT. In addition, RIP assays also revealed that LncDACH1 did not bind directly to HSP90, suggesting that LncDACH1 may regulate HSP90 expression levels through an indirect mechanism.

Previous studies have shown that LncRNA-protein complexes play an important role in the pathogenesis of several diseases[29]. Therefore, we hypothesized that LncDACH1 plays a role in NIH through this pathway. SRPK1 is a protein kinase that specifically phosphorylates proteins containing serine/arginine-rich domains and regulates a variety of RNA processing pathways including RNA stability, alternative splicing, and translation[30]. SRPK1 also regulates cell proliferation, apoptosis, and invasion in a variety of tumor cell types, thereby influencing oncogenesis[31]. We verified the interaction between full-length LncDACH1 and SRPK1 protein using RNA pulldown and RIP assays. We then predicted the specific binding site with catRAPID, designed truncated LncDACH1 fragments, and validated them with RNA pull-down assays. We found that one LncDACH1 fragment (nucleotides 774-1251), which is highly conserved in humans and mice, bound to SRPK1. Subsequently, we silenced LncDACH1 in dedifferentiated VSMCs and observed an upregulation of SRPK1. Conversely, overexpression of LncDACH1 downregulated SRPK1. Previous studies have found that SRPK1 regulates AKT activity through the AKT phosphatase PHLPP[32]. Subsequently, we demonstrated that SRPK1 modulates LncDACH1-mediated VSMC proliferation, migration, and phenotypic switching by regulating AKT activity. By simultaneously silencing LncDACH1 and SRPK1, we found that SRPK1 was able to partially reverse the promoting effect of LncDACH1 on p-AKT.

In the present study, we found that HSP90 mediated the nuclear translocation of SRPK1 by LncDACH1. The regulatory relationship between LncDACH1 and HSP90/SRPK1 will be discussed further here. Previous studies have shown that the interaction between HSP90 and SRPK1 in the cytoplasmic is crucial for inhibiting SRPK1 nuclear translocation[33]. In the present study, we found that silencing LncDACH1 enhanced the binding ability between HSP90 and SRPK1 through Co-IP experiments. This also explains the reason for silencing LncDACH1 to inhibit SRPK1 nuclear translocation. We then explored why LncDACH1 regulates the binding capacity between HSP90 and SRPK1. Our previous studies demonstrated that LncDACH1 could bind directly to SRPK1 but not to HSP90, and we proposed the hypothesis that LncDACH1 may bind to SRPK1 in competition with HSP90. In differentiated VSMCs, the binding of SRPK1 by LncDACH1 and HSP90 remained relatively balanced. In dedifferentiated VSMCs, however, LncDACH1 expression was downregulated, exposing the binding site of SRPK1 to HSP90 and resulting in enhanced binding of HSP90 to SRPK1 and inhibition of nuclear translocation of SRPK1. At the same time, the accumulation of more SRPK1 in the cytoplasm could further promote the expression level of P-AKT. This conjecture also explains why LncDACH1 can

regulate the binding ability between HSP90 and SRPK1. At present, we have yet to verify whether the binding site between LncDACH1-SRPK1 is consistent with that between HSP90-SRPK1, and a more rigorous experimental design is required to support this hypothesis in the future.

KLF9 is a member of the SP/KLF transcription factor family and binds to target gene promoters, enhancers, or silencers via three C-terminal C2H2 zinc fingers. KLF9 plays an important role in physiological processes such as cell growth, differentiation, proliferation, apoptosis, and metabolism, as well as multi-organ system development during embryogenesis[34]. Based on the results of UCSC and JASPAR prediction analyses, we verified that KLF9 transcriptionally regulates the LncDACH1 promoter. Specifically, mRNA expression levels of KLF9 were downregulated during VSMC phenotypic switching, consistent with the observed trends in LncDACH1. Interestingly, we found that overexpression of KLF9 in dedifferentiated VSMCs led to LncDACH1 upregulation, whereas KLF9 silencing resulted in reduced LncDACH1 expression. We then used luciferase activity and ChIP-qPCR assays to confirm that KLF9 directly binds to the LncDACH1 promoter and positively regulates its transcription. This result also explains the downregulation of LncDACH1 expression observed during VSMC phenotypic switching.

In vivo, we demonstrated that LncDACH1 regulated the extent of AVF NIH by constructing LncDACH1 CKO and AAV-LncDACH1 overexpressing mice. The extent of NIH increased after CKO of LncDACH1 and decreased after LncDACH1 overexpression. In vivo experiments were performed to verify the observed trends in SRPK1 and HSP90 expression; these results were consistent with those of the corresponding in vitro experiments. These experiments confirmed that LncDACH1 regulates NIH in mice with AVF. Subsequently, we validated the effect of modulating LncDACH1 on relevant phenotypic markers during NIH in AVF. We found that the expression level of the endothelial cell phenotype marker CD31 was unaffected in either CKO-LncDACH1 or AAV-LncDACH1, suggesting that the regulation of NIH by LncDACH1 may not be realized through endothelial cells; for the smooth muscle cell phenotype markers, we found that the expression of SM22α was downregulated in CKO-LncDACH1 whereas the expression of α-SMA, Vimentin and OPN expression was upregulatedthe opposite after overexpression of LncDACH1. A point worth noting here is that α-SMA is commonly used as a differentiation phenotype marker for VSMC in in vitro to detect the differentiation level of VSMC. Whereas in AVF, α-SMA is not only a differentiation phenotype marker for VSMC, it is also one of the phenotype markers for myofibroblasts. During NIH formation in AVF, fibroblasts distributed in the vascular adventitia are converted into myofibroblasts and thus promote the formation of NIH21. This suggests that in AVF, changes in α-SMA expression levels do not directly reflect the differentiation level of VSMC. In contrast, the down-regulation of SM22α, another differentiation phenotype marker, and the up-regulation of Vimentin and OPN, a dedifferentiation phenotype marker, suggest that the down-regulation of LncDACH1 can promote NIH formation by facilitating the phenotype switching of VSMC in the course of AVF NIH. The formation of NIH in AVF is a complex pathophysiological process, which consists of the

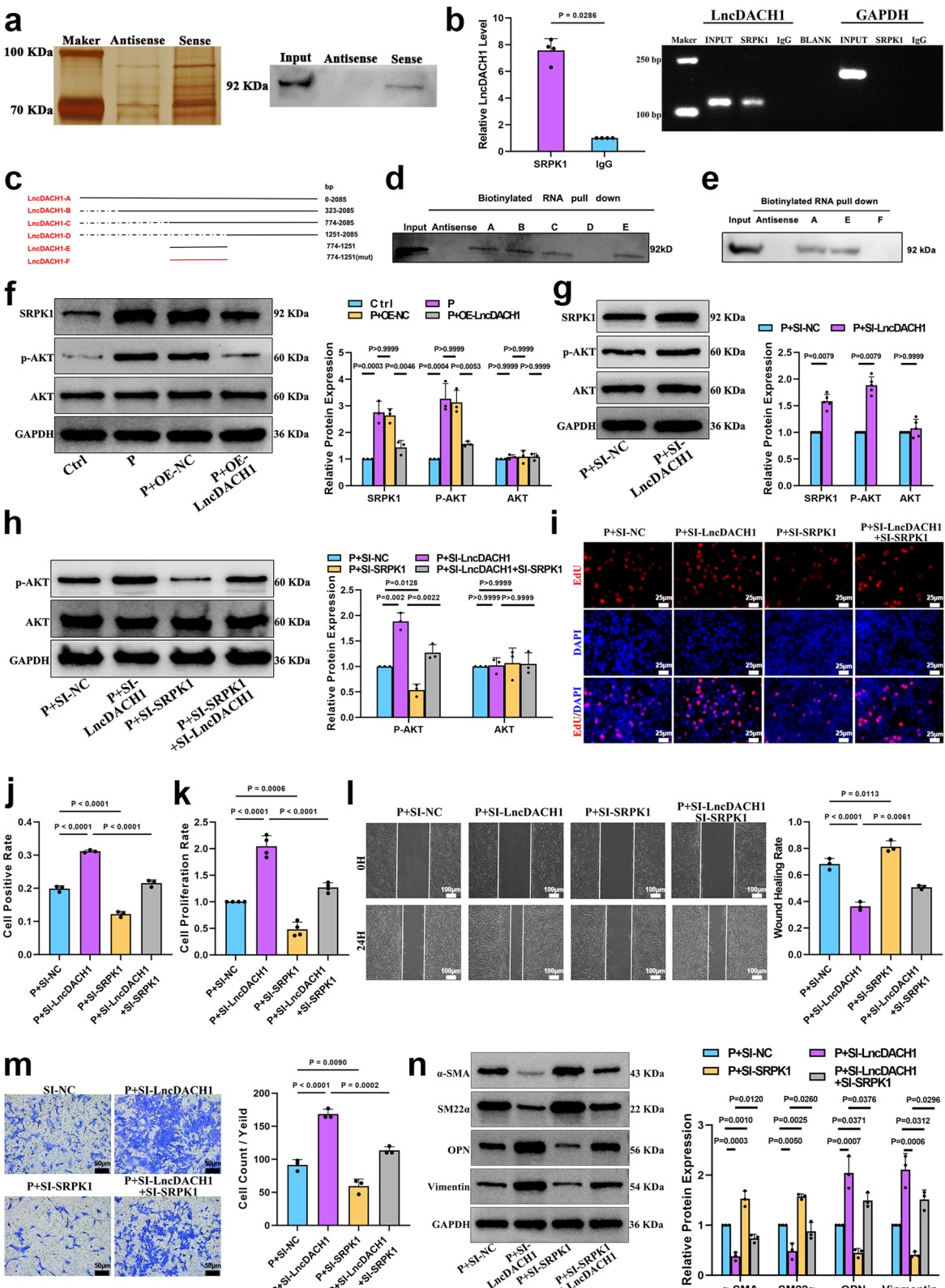

endothelial, fibroblastic, smooth muscle cell, and myofibroblastic cells, which induce, synergize, superimpose, or antagonize each other to form a highly complex network of actions21. In the present study, we observed up-regulation of the expression of the macrophage phenotype marker CD68 and the fibroblast phenotype marker FSP-1 in CKO-LncDACH1. We speculate that LncDACH1 regulates NIH not only by altering the differentiation level of VSMC but also

that the possible mechanism of action is that after VSMC dedifferentiation, a portion of the dedifferentiated VSMC have cellular interactions with fibroblasts or macrophages, which ultimately promotes the formation of NIH by regulating fibroblasts or macrophages.

Previous studies have demonstrated that various molecules play essential roles as targets in the formation of NIH, and various

**Fig. 5 | LncDACH1 promotes proliferation, migration and phenotype switching during VSMC dedifferentiation by binding to SRPK1. a** RNA Pull Down-LC/MS and Western Blot experiments were used to identify SRPK1 as one of the proteins pulled down by LncDACH1. **b** RIP assay was used to determine that LncDACH1 was pulled down by anti-SRPK1 antibody ($n = 4$). **c** Construction of LncDACH1 fragment (LncDACH1-A-F). fragment A, 0-2085 nt; fragment B, 323-2085 nt; fragment C, 774-2085 nt; fragment D, 1251-2085 nt; fragment E, 774-1251 nt; fragment F, 774-1251[mut] nt. **d** RNA Pull Down and Western Blot assays were used to determine the binding of LncDACH1 fragments A-E to SRPK1 ($n = 3$). **e** RNA Pull Down and Western Blot assays were used to determine the binding of LncDACH1 fragment F to SRPK1 ($n = 3$). **f, g** The protein expression levels of SRPK1, p-AKT and AKT after over-expression ($n = 3$) or silencing ($n = 5$) of LncDACH1 in VSMC were detected using Western Blot. VSMC is divided into the following four groups: si-NC, si-LncDACH1,

si-SRPK1, si-LncDACH1+si-SRPK1. **h** p-AKT and AKT protein expression levels were detected using Western Blot ($n = 3$). **i, j** The proliferation capacity of VSMC was tested by EdU staining assay ($n = 3$). Scale bar, 25 µm. **k** The proliferation capacity of VSMC was tested by CCK8 assay ($n = 4$). **l** The ability of VSMC to migrate was tested by Wound Healing assays ($n = 3$). Scale bar, 100 µm. **m** The ability of VSMC to migrate was tested by Transwell assay ($n = 3$). Scale bar, 50 µm. **n** Protein expression levels of VSMC differentiation phenotype markers and dedifferentiation phenotype markers were measured by Western Blot assay ($n = 3$). PDGF-BB (P), control (CTRL), negative control (NC), overexpression (OE), small interfering (SI). The $n$ numbers represent biologically independent samples. Data are presented as mean values ± SD (**b, f–h, j–n**). $P$-values were determined by two-sided nonparametric tests (**b, g**) and two-sided one-way ANOVA (**f, h, j–n**) by Bonferroni's multiple comparisons test. Source data are provided as a Source Data file.

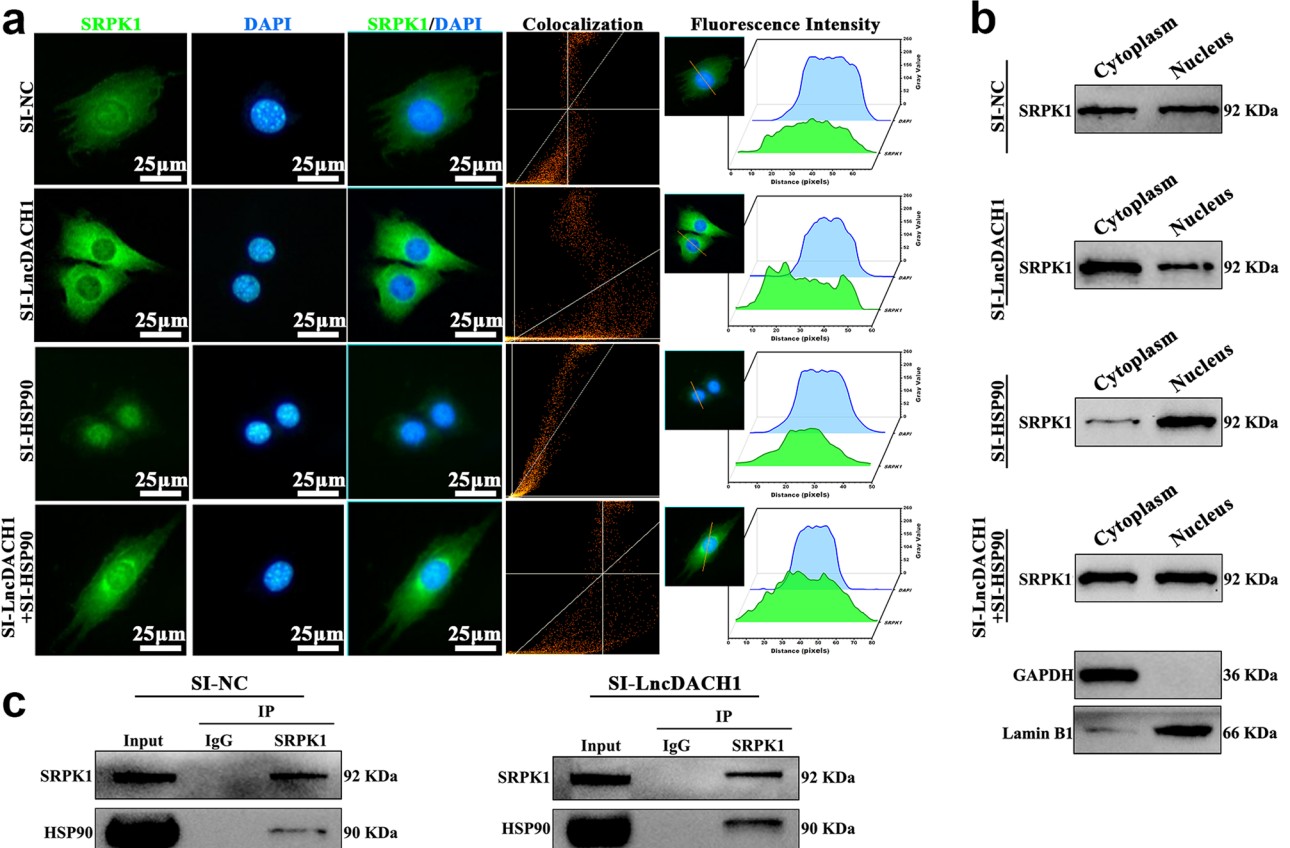

**Fig. 6 | HSP90 mediates LncDACH1-induced nuclear translocation of SRPK1. a** The distribution of SRPK1 in VSMC was examined using cellular immunofluorescence assays ($n = 3$). Scale bar, 25 µm. **b** Protein expression levels of SRPK1 in the nuclear cytoplasm of VSMC cells were examined using Western Blot assays ($n = 3$). **c** The binding ability of SRPK1 and HSP90 in VSMC using Co-IP assay ($n = 3$). Negative control (NC), small interfering (SI). The $n$ numbers represent biologically independent samples. Source data are provided as a Source Data file.

related target gene therapy studies have been conducted. However, the specific roles and therapeutic strategies of LncRNAs in forming AVF NIH have yet to be investigated. In this study, we verified that LncRNA-LncDACH1 plays an essential role in the process of AVF NIH, which implies that this class of molecules, LncRNAs, may be crucial in forming AVF NIH. This study may provide an entirely new type of target gene for treating AVF NIH.

Our study should be interpreted in view of its limitations. Animals with normal renal function may limit the utility of our experiments. Our study concludes that LncDACH1 regulates NIH formation through pro-proliferative and migratory effects on VSMCs in mice with normal kidney function. One of the mechanisms by which uremic toxins would further exacerbate NIH formation is the

elevated expression levels of potent mitogens such as PDGF. Therefore, LncDACH1 may further exert its regulatory role on NIH formation in this enhancement mechanism mediated by uremic toxins. Another limitation is that using only male mice is one of the limitations of this study. Previous studies have shown that sex differences are an independent risk factor for AVF maturation, the reason for which may be related to estrogen levels[35–37]. The main objective of this study was to investigate the underlying mechanism of action of the novel LncRNA LncDACH1 in AVF maturation. To ensure the rigor of the experimental results, we standardized the sex of the animals in this study. We used only male mice to avoid the potential influence of sex differences. However, our study does in no way exclude the possibility that LncDACH1 may play different roles

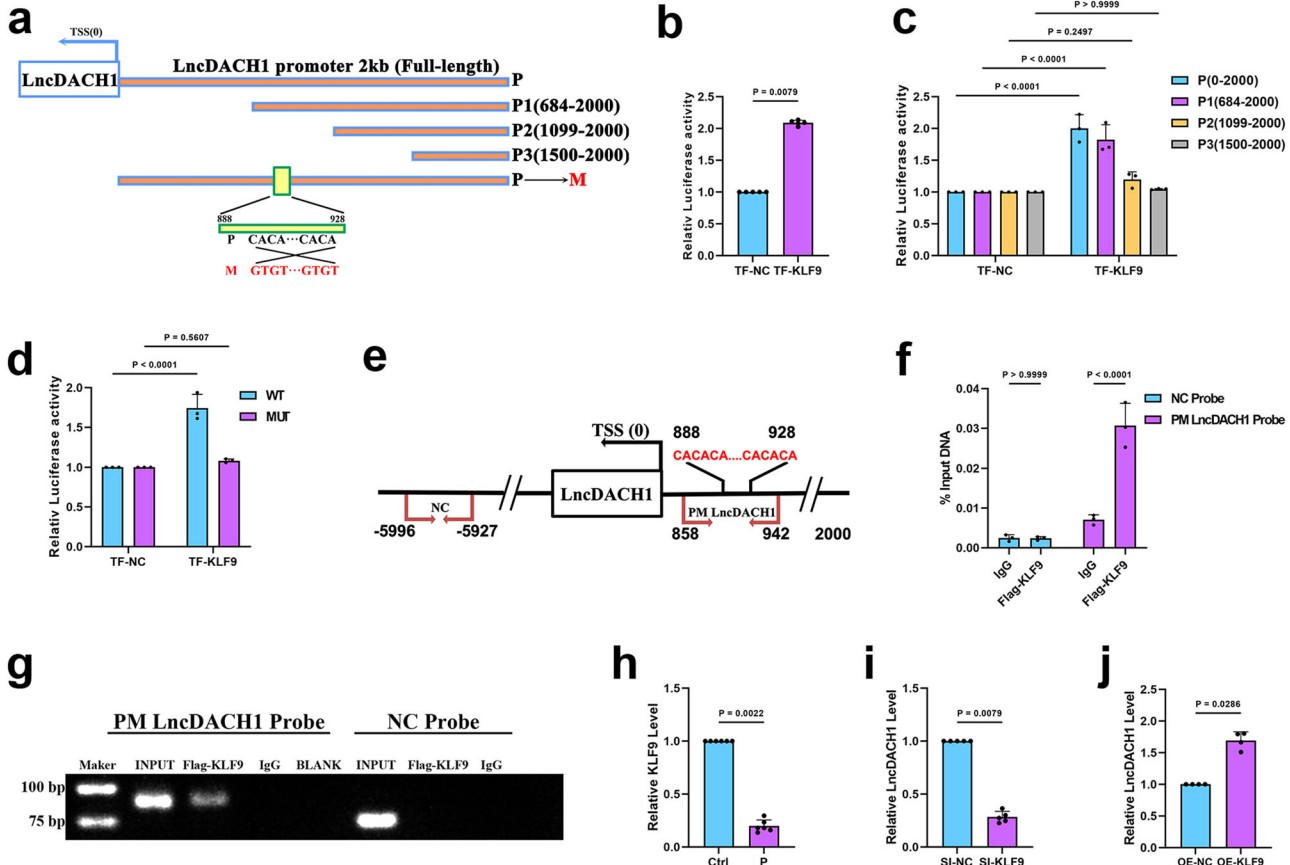

**Fig. 7 | KLF9 positively regulates LncDACH1 transcription. a** The figure shows the relative positions of the full length (FL) and truncated fragments (P1, P2, P3) of the LncDACH1 promoter sequence (2 kb) reporter gene and the mutation site (M). P1, 684-2000 nt; P2, 1099−2000 nt; P3, 1500−2000 nt; M, 888−928 mut. **b** Binding of the full-length (FL) LncDACH1 promoter sequence (2 kb) to the transcription factor KLF9 was determined using luciferase activity assay (*n* = 5). **c** Binding of the LncDACH1 promoter truncation fragments (P1, P2, P3) to the transcription factor KLF9 was determined using luciferase activity assay (*n* = 3). **d** Binding of the M mutant vector to the transcription factor KLF9 was determined using luciferase activity assay (*n* = 3). **e** The figure shows the relative positions of the qPCR negative control probe and the predicted binding fragment probe in the ChIP-qPCR experiment. **f, g** Binding of the LncDACH1 promoter sequence to KLF9 was examined using ChIP-qPCR assay (*n* = 3). **h** Changes in mRNA expression of KLF9 during VSMC dedifferentiation using qRT-PCR (*n* = 6). **i** LncDACH1 expression levels were measured by qRT-PCR after transfection of si-KLF9 or scrambled siRNA in VSMC (*n* = 5). **j** LncDACH1 expression levels were measured by qRT-PCR after transfection of KLF9- pcDNA3.0 or pcDNA3.0-Vector in VSMC (*n* = 4). Promoter (P), mutant (M), transcription factor (TF), transcription start site (TSS), control (CTRL), negative control (NC), overexpression (OE), small interfering (SI). The *n* numbers represent biologically independent samples. Data are presented as mean values ± SD (**b**–**d**, **f**, **h**–**j**). *P*-values were determined by two-sided nonparametric tests (**b**, **h**–**j**) and two-sided 2way ANOVA (**c**, **d**, **f**) by Bonferroni's multiple comparisons test. Source data are provided as a Source Data file.

in AVF maturation in mice of different sexes. This issue merits future studies for clarification. In addition, although we confirmed that LncDACH1 downregulation promotes HSP90 and SRPK1 expression, we have not identified the pathway through which it exerts this effect. Previous studies have shown that LncRNA regulates post-translational protein modifications through pathways such as ubiquitination, phosphorylation, acetylation, and autophagy, all of which affect protein expression levels and activity[29]. We hypothesize that LncDACH1 influences protein expression levels by regulating post-translational modifications; thus, more comprehensive studies need to be conducted to confirm this hypothesis.

In summary, our study demonstrates that LncDACH1 plays an important role in AVF NIH. Mechanistically, we identified the KLF9-LncDACH1-HSP90/SRPK1-AKT signaling axis, where KLF9 was downregulated in AVF NIH, leading to downregulation of LncDACH1, which in turn promoted SRPK1 and HSP90 to reactivate p-AKT. LncDACH1 also inhibited SRPK1 nuclear translocation may through enhancing the binding ability between HSP90 and SRPK1 (Fig. 10). These findings provide new insights into the molecular mechanisms

underlying AVF NIH and suggest that LncDACH1 may be a potential target for the prevention or treatment of AVF NIH.

## Methods

### Ethics statement

**This research complies with all relevant ethical regulations**

**Human AVF sample collection.** Our studies were approved by the Ethics Committee of the Second Affiliated Hospital of Harbin Medical University. We informed all related patients of the use of these specimens and got the written informed consent. Preoperative AVF veins and stenosis AVF veins were collected from different patients. Preoperative AVF veins were collected from uraemic patients who were about to undergo AVF surgery, whereas stenosis AVF veins were collected from veins that were discarded during revision or reconstruction of focal stenosis surgery for AVF treatment. The patient demographics and characteristics are described in Supplementary Table 1. Due to the current availability of endovascular interventions, there are fewer patients with focal endovascular fistula revisions or reconstructions than in previous years, making it

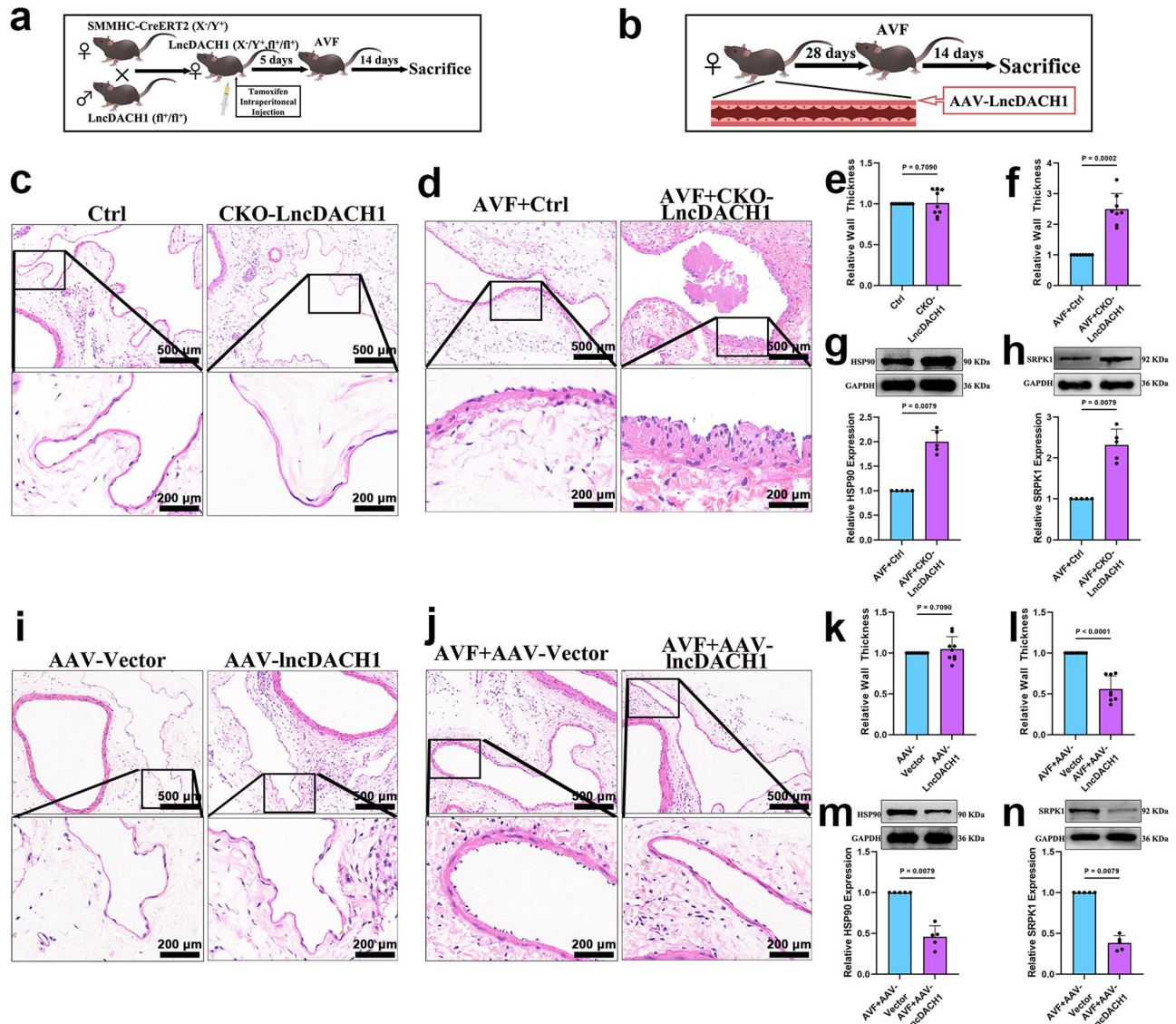

**Fig. 8 | LncDACH1 mediates NIH in AVF mouse model. a** The diagram shows the construction strategy of smooth muscle cell-specific LncDACH1 conditional knockout mouse. **b** The diagram shows the construction strategy of adeno-associated Virus overexpressing LncDACH1 mouse. **c**, **e** HE staining of vascular tissue and morphometric analysis of the extent of vascular neointimal hyperplasia in mouse without established AVF in CTRL ($n = 9$) and CKO-LncDACH1 ($n = 9$), respectively. **d**, **f** HE staining of vascular tissue and morphometric analysis of the extent of vascular neointimal hyperplasia in mouse with established AVF in CTRL ($n = 8$) and CKO-LncDACH1 ($n = 8$), respectively. **g**, **h** The expression levels of SRPK1 and HSP90 proteins in vascular tissues of CTRL and CKO-LncDACH1 mouse with established AVF were examined using Western Blot assays ($n = 5$ at each group). **i**, **k** HE staining of vascular tissue and morphometric analysis of the extent of vascular neointimal hyperplasia in mouse without established AVF in AAV-Vector ($n = 9$) and AAV-LncDACH1 ($n = 9$), respectively. **j**, **l** HE staining of vascular tissue and morphometric analysis of the extent of vascular neointimal hyperplasia in mouse with established AVF in AAV-Vector ($n = 9$) and AAV-LncDACH1 ($n = 8$), respectively. **m**, **n** The expression levels of SRPK1 and HSP90 proteins in vascular tissues of AAV-Vector ($n = 5$) and AAV-LncDACH1 ($n = 5$) mouse with established AVF were examined using Western Blot assays. Arteriovenous fistula (AVF), conditional knockout (CKO), adeno-associated virus (AAV), control (CTRL). The $n$ numbers represent biologically independent samples. Data are presented as mean values ± SD (**e**–**h**, **k**–**n**). *P*-values were determined by two-sided nonparametric tests (**e**–**h**, **k**–**n**). Source data are provided as a Source Data file.

relatively difficult to obtain veins from this experimental group; therefore, only $n = 4$ at each group human specimens were available for analysis.

### Animal model

In this study, male wildtype C57BL/6 J mice at the age of 6 to 8 weeks (Animal Center of the Second Affiliated Hospital of Harbin Medical University, Harbin, China) were used. The research was approved by the Ethics Committee on Use and Care of Animal Center of Harbin Medical University. They were fed in the standard room without pathogens, and the light cycle (12 h light to 12 h dark), humidity (50%

± 5%) and temperature (20 °C to 22 °C) were specifically controlled. Isoflurane (2–3%) was used for anesthesia, and the aorta was punctured with a 25-gauge needle to establish an aortocaval fistula. The success of AVF establishment was confirmed by observing alternating arterial and venous flow in the IVC. Pain was controlled 48 h postoperatively with an intraperitoneal injection of 0.1 mg/kg buprenorphine. Mice were euthanized and AVF veins were collected on postoperative Days 0, 7, 14, and 21. Since we need to conduct animal experiments based on the successfully establishment of the AVFs, only successfully establishment AVFs were used in this study, and unsuccessfully establishment AVFs due to surgical technique

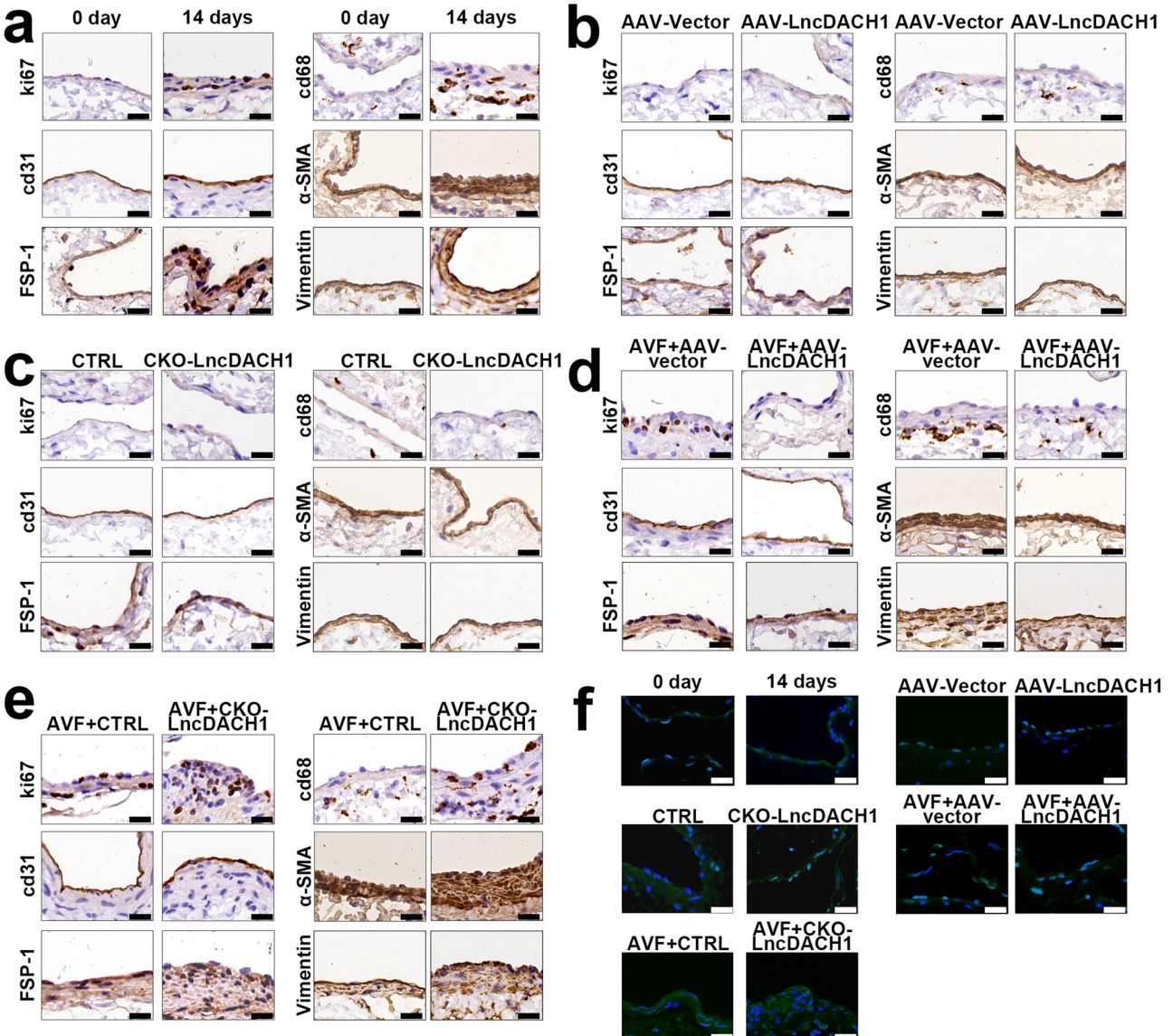

**Fig. 9 | LncDACH1 affected NIH in the context of AVF by regulating VSMC phenotype switching. a** The expression level of Ki67, CD68, CD31, α-SMA, Vimentin and FSP-1 in mouse before AVF (*n* = 10) versus 14 days after AVF (*n* = 9) were detected by immunohistochemistry assay. **b** The expression level of Ki67, CD68, CD31, α-SMA, Vimentin and FSP-1 in mouse without established AVF in AAV-Vector (*n* = 9) and AAV-LncDACH1 (*n* = 9) were detected by immunohistochemistry assay. **c** The expression level of Ki67, CD68, CD31, α-SMA, Vimentin and FSP-1 in mouse without established AVF in CTRL (*n* = 9) and CKO-LncDACH1 (*n* = 9) were detected by immunohistochemistry assay. **d** The expression level of Ki67, CD68,

CD31, α-SMA, Vimentin and FSP-1 in mouse with established AVF in AAV-Vector (*n* = 9) and AAV-LncDACH1 (*n* = 8) were detected by immunohistochemistry assay. **e** The expression level of Ki67, CD68, CD31, α-SMA, Vimentin and FSP-1 in mouse with established AVF in CTRL (*n* = 8) and CKO-LncDACH1 (*n* = 8) were detected by immunohistochemistry assay. **f** The expression levels of Tunel in the above groups by immunofluorescence assay. Arteriovenous fistula (AVF), conditional knockout (CKO), adeno-associated virus (AAV), control (CTRL). The statistical analysis is shown in Supplementary Fig. 10. The *n* numbers represent biologically independent samples. Scale bar, 20 μm. Source data are provided as a Source Data file.

and death within 24 h after surgery, etc. were excluded (Supplementary Table 2).

### Generation of smooth muscle cell-specific conditional knockout LncDACH1 mice

SMMHC-CreERT2 mice were purchased from Cyagen Biosciences Inc. (USA). LncDACH1fl/fl mice were constructed by Biocytogen Co. (China). To obtain smooth muscle cell-specific conditional LncDACH1-knockout mice, LncDACH1 (flox$^+$/flox$^+$, Cre$^-$) mice were hybridization with Cre transgenic mice 80 mg/kg body weight tamoxifen (Sigma) was injected into the intraperitoneal of adult LncDACH1 (flox+/flox+, Cre+) mice once a day, for five days. The control group of conditional knockout mice included LncDACH1 (flox+/flox+, Cre-) mice that lacked

the Cre transgene were administered tamoxifen injections. Male LncDACH1 (flox$^+$/flox$^+$, Cre$^-$) mice and LncDACH1 (flox+/flox+, Cre-) mice at the age of 6 to 8 weeks were used. RT–PCR was used to analyse tail genomic DNA to verify the genotype of the LncDACH1 (flox/flox) and Cre mice. The LncDACH1 (flox/flox) primer pair in our study: loxP-F: CCAGAAGCACCCAGGACATTGTTGT and loxP-R: ACATCACA-GAGCCACTGTAAGGAGTT. The length of the product that is amplified from mutant mice is expected to be 394 bp, and the length of the product that is amplified from wild-type mice is expected to be 306 bp.

The specific primers used to identify Cre mice for PCR genotyping include primers specific for SMMHC-CreERT2: F1: 5′-TGACCC-CATCTCTTCACTCC-3′ and R1: 5′-AGTCCCTCACATCCTCAGGTT-3′. The product length for CRE mice is expected to be 287 bp, and no

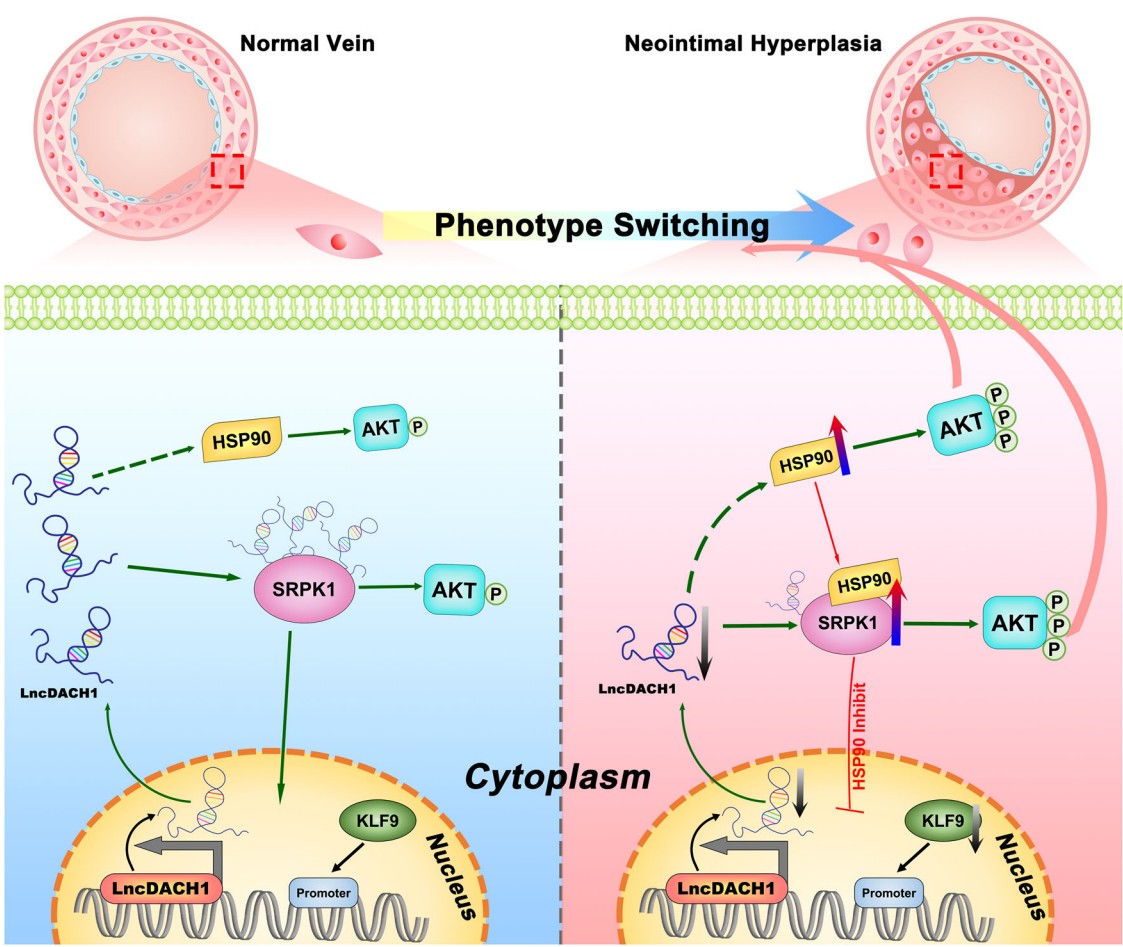

**Fig. 10 | Mechanism of action of LncDACH1 mediated AVF NIH.** KLF9 was downregulated in AVF NIH, leading to downregulation of LncDACH1, which in turn promoted SRPK1 and HSP90 to reactivate p-AKT. LncDACH1 inhibited SRPK1 nuclear translocation may through enhancing the binding ability between HSP90 and SRPK1.

product is expected for wild-type mice. The primers of wild-type: F2: 5′-CAGCCAACTTTACGCCTAGC-3′ and R2: 5′-TCTCAAGATGGACCTAA-TACGG-3′. No product was expected for the CRE mice, and a product length of 180 bp was expected for the wild-type mice. The degree to which LncDACH1 expression was decreased in the blood vessels of the LncDACH1-CKO mice was measured by qRT–PCR.

### Generation of adeno-associated virus overexpressing LncDACH1 mice

An adeno-associated virus vector carrying LncDACH1 (OE-LncDACH1) was constructed by GeneChem (China). Mix adeno-associated virus (10 μl; $1.4 \times 10^{12}$ vg/ml) with biogel PluronicF-127 (50ul) in the proportion and then topically applied to the surface of the vessels for 5 min; then, the gel was allowed to solidify. Four weeks later, the vessel tissues were isolated, and transfection efficiency was verified by qRT-PCR.

### Histology

Human and mice tissues were first rinsed in PBS and then fixed in 4% PFA. Tissues were embedded in paraffin wax and the section thickness were 5μm. For mice with established AVF, near the point of vessel puncture (approximately 1 mm in length) was sectioned at 5μm intervals (approximately 100-120 sections per sample). The section with the greatest neointima area as identified by bright field microscopy. The morphology was observed with hematoxylin and eosin (HE) staining. The mean neointimal hyperplasia thickness was measured for each sample by using ImageJ software.

### Immunofluorescence (IF)

After fixed in 4% paraformaldehyde for 15 min, cells then permeabilized by 0.2% Triton-100 PBS for 20 min. After PBS washed for 5 times, blocked the cells in 5% BSA for 1 h at the room temperature (RT), and then incubated in the primary antibody at 4 °C overnight. After five washes with PBS, cells were incubated with the corresponding fluorescent secondary antibody for 8 h RT. Finally, nuclei were stained with DAPI for 15 min. The results were visualized by fluorescence microscopy (Leica).

### Cell culture and treatments

The HEK-293T cell line (Otwobiotech Technology, HTX1559) and mouse (Otwobiotech Technology, HTX1886) and human VSMC cell lines (Otwobiotech Technology, HTX2352) were purchased from Otwobiotech Technology (China). All the cells were cultured in medium (Gibco) supplemented with 10% FBS (Corning) at 37 °C containing 5% $CO_2$. To induce VSMC migration, proliferation, and phenotype switching, we treated VSMCs with different concentrations (0, 5, 10, and 20 ng/mL) of PDGF-BB (Peprotech) for 48 h. Subsequently, VSMCs were treated with PDGF-BB (10 ng/mL) at different times (24, 48, and 72 h) and then analyzed to measure LncDACH1 expression.

### Preparation of plasmid expression vector and siRNA

LncDACH1-siRNA, KLF9-siRNA, SRPK1-siRNA, HSP90-siRNA and scrambled siRNA were constructed by GenePharma (China); The LncDACH1-A (nucleotides 0–2085) -pcDNA3.0 vector, LncDACH1-B

(nucleotides 323–2085) -pcDNA3.0 vector, LncDACH1-C (nucleotides 774–2085) -pcDNA3.0 vector, LncDACH1-D (nucleotides 1251–2085) -pcDNA3.0 vector, LncDACH1-E (nucleotides 774–1251) -pcDNA3.0 vector, LncDACH1-F (nucleotides 774–1251 mutant [MUT]) -pcDNA3.0 vector, KLF9-pcDNA3.0 vector, Flag-KLF9-pcDNA3.0 and pcDNA3.0 empty vector were constructed by GeneChem (China). The LncDACH1 promoter-P (0-2000), LncDACH1 promoter-P1 (684-2000), LncDACH1 promoter-P2 (1099-2000), LncDACH1 promoter-P3 (1500-2000), LncDACH1 promoter-P (0-2000 mutant [MUT]) was cloned into the luciferase reporter GV238 vector (GeneChem). Transfection of siRNA or pcDNA into VSMCs was performed with Lipo2000 (Invitrogen). LncDACH1-siRNA were transfected into cells for LncDACH1 knockdown at a final concentration of 100 nM. HSP90-siRNA were transfected into cells for HSP90 knockdown at a final concentration of 80 nM. SRPK1-siRNA were transfected into cells for SRPK1 knockdown at a final concentration of 60 nM. KLF9-siRNA was transfected into cells for KLF9 knockdown at a final concentration of 60 nM. Scrambled siRNA was transfected into cells for negative control at the same final concentrations as in the corresponding experimental groups. The relevant siRNA sequences are presented in the Supplementary Material (Supplementary Table 3).

### RNA isolation, reverse transcription and quantitative real-time PCR (qRT-PCR)

Both cells and tissue samples Total RNA were extracted by TRIzol reagent (Invitrogen). cDNA synthesis using a reverse transcription kit (Takara). Quantitative real-time PCR was performed using AceQ Universal SYBR qPCR Master Mix (Vazyme). Data were analyzed by ΔΔ Ct method. GAPDH as internal reference gene. PCR primer sequences were shown in the Supplementary Material (Supplementary Table 4).

### Co-Immunoprecipitation (Co-IP)

VSMC were washed three times with PBS and fully lysed on ice with cell lysis buffer containing protease inhibitors for 30 min. The lysates were centrifuged at 11,000 $g$ for 15 min at 4 °C, and the supernatants were collected. SRPK1 antibody (5ug) or control mouse immunoglobulin G (IgG) (5ug) was added to the protein samples and incubated at 4 °C overnight. The prepared Protein A/G-MagBeads suspension was added to the protein samples and shaken in a vertical mixer at 4 °C for 2 h. The protein samples were denatured and eluted in an eluent in a boiling water bath for 10 min. Magnetic beads were adsorbed using a magnetic holder, and the sample supernatant was collected for the next step of the experiment.

### Western blot analysis

The equal mass proteins were separated on SDS polyacrylamide gels and transferred to polyvinylidene fluoride (PVDF) membranes. The membranes were then incubated in primary antibody at 4 °C overnight. The western blot bands were imaged using a Biochem System (BIO-RAD). Primary antibodies in our study were antibodies against α-SMA (1:1000, Abcam, ab124964), Sm22α (1:1000, Abcam, ab14106), Opn (1:1000, Abcam, ab283656), Vimentin (1:1000, Abcam, ab92547), P-AKT (1:2000, Cst, 4060), AKT (1:1000, Cst, 4685), SRPK1 (1:1000, ProteinTech, 14073-1-AP), HSP90 (1:1000, Santa Cruz, sc-13119), FLAG (1:50, Cst, 14793), KLF9 (1:200, Santa Cruz, sc-376422), Lamin B1 (1:20000, ProteinTech, 66095-1-Ig) and GAPDH (1:100000, ProteinTech, 60004-1-Ig).

### VSMC proliferation

VSMCs were cultured in 96-well plates. Then added 10 μl cell counting kit-8 (CCK-8 assay) (Yeasen) each well and incubated for 30 min. Absorbance was measured using a multifunctional enzyme marker (Biotek). Similarly, cell proliferation was measured with a 5-ethynyl-2′-deoxyuridine (EdU) kit (APExBIO). Images were obtained with a fluorescence-inverted microscope (Leica). The directions provided by the manufacturer were followed to conduct the experimental procedures.

### VSMC migration

In the wound healing assay, VSMCs were seeded in 6-well plates. Uniform linear scratches were created with the tip of a 100 μl pipette, and images were captured at various times using a Leica microscope. In transwell experiments, VSMCs (200 μl of serum-free DMEM/well) were added to the 24-well plate upper chamber, and 600 μl of DMEM supplemented with 10% FBS added in the lower chamber. The cells were incubated for the indicated times, and the cells in the lower chamber were fixed with 4% PFA and stained with crystalline violet. The cells in the upper chamber were then removed with a cotton swab, and images were captured by a Leica microscope.

### Fluorescent in situ hybridization (FISH)

Experiments were performed using a Fluorescent In Situ Hybridization Kit (RiboBio). Cells were fixed in 4% PFA for 20 min, and the slides were dried at room temperature (RT). The LncDACH1-specific probe was incubated in the cells for 18 h at 37 °C in hybridization solution. After hybridization, the slides were washed six times with prewarmed wash buffer and finally stained with DAPI. Finally, images were obtained by fluorescence inverted microscope (Leica).

### Cytoplasmic and nuclear RNA/protein purification

Cytoplasmic and nuclear RNA isolation was used by Cytoplasmic & Nuclear RNA Purification Kit (Norgen Biotek). 200 μl of cell lysis buffer was added, and the cells were lysed and transferred to a centrifuge tube. After centrifugation, the nuclei could be visualized at the bottom of the tube, and the supernatant containing cytoplasmic RNA was transferred to a new centrifuge tube. The RNA was extracted by adding the appropriate amounts of buffer and anhydrous ethanol to the RNA purification column.

Cytoplasmic and nuclear proteins were isolated by Cytoplasmic & Nuclear Protein Purification Kit (Beyotime Biotechnology). Cells were incubation to ice bath for 15 min with PMSF. Then samples were centrifuged 12,000 $g$ for 6 min at 4 °C. After centrifugation, nuclei proteins could be visualized at the bottom of the tube. The supernatant was quickly collected and transferred the cytoplasmic proteins to another centrifuge tubes.

### Luciferase activity assay

293 T cells were seeded in a 24-well plate with 500 μl complete medium and co-transfected with reporter gene plasmids and transcription factor plasmids. Cell lysis buffer was added 48 h after transfection, and luciferase activity was measured by luciferase activity assay Kit (Beyotime Biotechnology). The Renilla luciferase plasmid was used as an internal control. Chemiluminescence was measured using a multi-functional enzyme marker (Biotek).

### Mass spectrometry assay

The total number of samples analyzed was 6 groups, with $n = 3$ in the SI-NC group and $n = 3$ in the SI-LncDACH1 group. The n numbers represent biologically independent samples. Take 100 μg of protein and dissolve to ~1 μg/μL using U2 lysis buffer add 5x volume of 100 mM TEAB to dilute the protein 6-fold. Add 1.2 μL of 0.5 M CaCl2 and centrifuged with shaking. Add trypsin (trypsin: protein = 1:100) and incubate at 37 °C for more than 8 h. Weigh 10 mg of C18 column stock, corresponding to each 100 μg of peptide sample. The samples were washed twice with 0.1% FA + 3% ACN for desalting and eluted with 1 mL of 0.1% FA + 80% ACN. Peptide samples were dissolved with 20 μL dissolution buffer (0.5 M TEAB), 70 μL isopropanol was added, and centrifuged with shaking. Labeling was performed according to the instructions of the iTRAQ-8 Labeling Kit (SCIEX), and then the mixed peptides were subjected to hierarchical separation of peptide samples

by applying an Ultimate 3000 HPLC system (Thermo DINOEX). Mass spectrometry data were acquired using a Triple TOF 5600 + Liquid Mass Spectrometry system (AB SCIEX). The mass spectrometry downstream data were analyzed by database search using ProteinPilot 4.5 software (AB Sciex).

### RNA pull-down
Biotinylated LncDACH1 sense or antisense RNA probes were synthesized. The nucleic acid-compatible streptavidin magnetic beads were washed by buffer for three times and then resuspended. Biotin-RNA was added to the beads and incubated. Finally, the protein was eluted using biotin elution buffer. Analyzed by mass spectrometry and Western blotting. The antisense gene of LncDACH1 was used as a negative control.

### RNA binding protein immunoprecipitation assay (RIP)
VSMCs were lysed with RIPA buffer and ribonuclease inhibitor (Solarbio) for 5 min. Subsequently, protein and magnetic beads were incubated with SRPK1 antibody, HSP90 antibody or control mouse immunoglobulin G (IgG) overnight at 4 °C. Immunoprecipitation of the complexes was performed, followed by RNA extraction with TRIzol, reverse transcription, qRT–PCR.

### Chromatin immunoprecipitation (ChIP) assay
Chromatin protein complexes were precipitated with an anti-Flag antibody or isotype-matched control IgG, and then, Protein A/G MagBeads were enriched. After washing three times, the bead-bound immune complexes were eluted with 200 μL of elution buffer. Tris-EDTA buffer was added to reverse the DNA–protein cross-linking. The DNA fragments were precipitated and purified and subjected to qRT–PCR.

### Statistical analysis
All the experiments were repeated at least three times, and the data were presented as the mean ± standard deviation (SD). GraphPad Prism 9. 3 software was used for statistical analysis of variance. Nonparametric test was used to compare significant differences between the two groups, and one-way or two-way ANOVA was used for more than two groups. A $P$ value < 0. 05 was considered statistically significant.

### Reporting summary
Further information on research design is available in the Nature Portfolio Reporting Summary linked to this article.

## Data availability
The mass spectrometry proteomics data generated in this study have been deposited in the ProteomeXchange Consortium via the iProX partner repository with the dataset identifier database[https://www.iprox.cn//page/project.html?id=IPX0005596000]. The full-length LncDACH1 Promoter sequence and the predicted binding sites of KLF9 were obtained through the UCSC database[https://genome.ucsc.edu/] and JASPAR database[https://jaspar.elixir.no/]. The predicted binding and sequence of LncDACH1 with SRPK1 was obtained through the RPISeq database[http://pridb.gdcb.iastate.edu/RPISeq/] and catRAPID database[http://s.tartaglialab.com/page/catrapid_group]. Source data are provided with this paper.

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

## Acknowledgements

This study was supported by the National Natural Science Foundation of China (82170756) and the Technology Research and Development Program of Heilongjiang (GY2019YF0165) to J.J. We thank J.C. of the Animal Center of the Second Affiliated Hospital of Harbin Medical University for technical support for the establishment of the animal model for this study. We also acknowledge H.ZD.'s significant contribution in providing statistical data for this study.

## Author contributions

J.J., C.Z. and Z.L. vised the study design, main conceptual ideas, and project outline; Z.L. contributed to cell experiments and draw mechanism diagrams; Y.Z. and Z.L. contributed to animal experiments; B.C. and Z.P. processed and analyzed the data; C.Z. procured human samples; Z.L. wrote the manuscript. All authors approved the manuscript.

## Competing interests

The authors declare no competing interests.
