## [Peer Review File · Nature Communications]

REVIEWER COMMENTS

Reviewer #1 (Remarks to the Author):

This is a study on a previously identified lncRNA lncDACH1 that was discovered by this team and now focusing on the pathology of AVF failure. There are mechanistic studies and experimental in vivo studies to prove both the mechanism and the functionality in this setting. There are some major limitations/areas to improve..

1. The authors state "Additional animal experiments were not performed to avoid wasting valuable animal resources when we observed statistically significant differences with smaller sample sizes." Formal power calculations are needed for in vivo experiments to ensure powered correctly in the design of the experiment. This is missing and must be based on the model.

2. The delivery of the AAV is not sufficiently addressed. It is done by qPCR but needs to be done with in situ localisation for the lncRNA, plus a control reporter gene AAV should be used to show location and duration of expression (GFP or lacZ, for example). This is essential to show the location of expression of the lncRNA to understand the mechanisms and functional effects.

3. In situ localisation for lncRNADACH1 is required to show endogenous levels during disease progression in vivo and in patient samples.

4. Nuclear/cytoplasmic expression levels need to be quantified with additional assays and controls for the C/N portions to ensure the techniques are applied correctly. This is also relevant to the siRNA - in that setting does that affect both N and C portions or just Cytoplasmic? this is important to understand the mechanism of action.

5. In the siRNA experiments (mixing), please be clear on the total doses of siRNA in each group.

6. Data in Figure 1 needs improving:

- Data points needed in bar plots for clarity of data
- in situ needed for co-localisation with vSMC markers to co-localise the lncRNA in tissue samples.
- Fig 1 H is poor - one cell shown only.
- As already stated 1H needs controls. e.g. NEAT1. and so on.

7. In figure 2,

- the plasmid over expression in Fig 2 a is very modest.
- Scale bars needed in images
- the cell proliferation rate appears to be very low in these experiments, what is this?
- the data in Fig 2 j looks very marginal. please be very clear on stats and replicates (biological).

8. Figure 3

- In A, what are the endogenous levels of the lncRNA? these graphs are all relative and we do not get a realistic assessment of the endogenous levels and thus, the endogenous levels that can be inhibited.

- Scale bars needed
- Similar additional comments to those in Figure 2 regarding data points and replicates.

9. In figure 4p, there are no controls at all. This is therefore a negative association and not a controlled experiment.

10. In Figure 5 M, again the data look marginal for the graphs in the green and yellow bars and is this not the important comparison?

11. In the in vivo experiments, more data are needed in the main figure to understand the effects being observed.

- The vessel morphometry in both experiments needs full analysis. Endothelial coverage, SMC layer markers and so on. Proliferation rates too, especially of the SMC layer. Lack of expression and gain of expression need confirming with in situ as well as the PCR. These data are interesting but incomplete. Finally, the mechanistic effects of the lncRNA are incompletely assessed. The western blots alone are not convincing and n=1 shown.

Reviewer #2 (Remarks to the Author):

Overall summary:

This paper studies the role of LncDACH1 in venous neointimal hyperplasia in mice with AVFs. The author uses cell culture experiments to investigate the role of LncDACH1 in multiple different in vitro experiments. The authors place AVFs in mice and show data for vascular remodeling. I have several questions for the authors with respects to the data that is presented:

Mice AVFs:

The authors need to show data for additional measures including neointima area, media + adventitia area, lumen vessel area (LVA), Ki-67, and TUNEL.

Also, they should show the other cellular phenotypes that are present in the vessel wall including fibroblast, SMC phenotypes (contractile vs synthetic), and endothelial cells, and macrophages.

Need n/m+a ratio, LVA, LVA/n

Cell density

The removal of mice that were only successful biases the data and we need to know what did the other animals staining features with respect to the readouts. How many animals were there? Also, the animals have normal kidney function and this limits the utility of the animal model.

Human AVF Sample Collection:

What are the demographics, age, comorbidities

Figures: OK

Reviewer #3 (Remarks to the Author):

This is a well done study that provides new information regarding the mechanisms of NIH in AVF. This study provides evidence of a potential intervention for NIH in AVF, namely the LncDACH1 pathway, which is significant due to the short-lived nature of AVF.

While the authors clearly indicate that more studies are needed to confirm these findings, the current study draws reasonable conclusions from the data generated.

As stated in the manuscript the this study is lacking in the number of human samples examined due to the scarcity of such tissue. The authors acknowledge this limitation.

The methodology is sound and for the most part the information supplied is sufficient for the work to be replicated. In line 132, the authors at the specific ratio is absent and needs to be stated.

The authors also state that animal studies were performed until significance was reached. This design is not standard and a power analysis should have been performed to determine animal number prior to initiation of the study.

Another omission is the statement of a clear hypothesis in the introduction. While the authors do refer to their hypothesis later in the text, the hypothesis needs to be stated upfront.

There is also a confusing statement in line 295 stating "LncDACH1 was more significantly downregulated in mouse VSMCs during dedifferentiation". Was this tested statistically or is it simply anecdotal observation of the data?

Line 612 has a typographical error

REVIEWER COMMENTS

Reviewer #1 (Remarks to the Author):

This is a study on a previously identified lncRNA LncDACH1 that was discovered by this team and now focusing on the pathology of AVF failure. There are mechanistic studies and experimental in vivo studies to prove both the mechanism and the functionality in this setting. There are some major limitations/areas to improve.

1. The authors state "Additional animal experiments were not performed to avoid wasting valuable animal resources when we observed statistically significant differences with smaller sample sizes." Formal power calculations are needed for in vivo experiments to ensure powered correctly in the design of the experiment. This is missing and must be based on the model.

Response: Thank you for your suggestions. We truly believe that the concern you raised is of paramount importance. We performed our power analysis separately for in vivo experiments using PASS 15 software, with an assumption of Alpha = 0.05 and Target Power > 0.90. With such a setting, only n=5 experimental animals per group are required for adequate and satisfactory statistical analyses. Yet, to further address your concern and ensure the accuracy of our experimental conclusions, we conducted additional in vivo experiments, adding up to a minimum of n=8 animals per group. The addition of the new data does not change our original conclusions.

0 days and 14 days

Alternative Hypothesis: $H1: \delta = \mu1 - \mu2 \neq 0$

Target Power	Actual Power	N1	N2	N	$\mu1$	$\mu2$	δ	σ	Alpha
0.90	0.91994	3	3	6	12.0	26.6	-14.5	3.9	0.050

Dropout-Inflated Sample Size

Dropout Rate	Sample Size			Dropout-Inflated Enrollment Sample Size			Expected Number of Dropouts		
	N1	N2	N	N1'	N2'	N'	D1	D2	D
10%	3	3	6	4	4	8	1	1	2

Design Tab

Solve For:	Sample Size
Alternative Hypothesis:	Two-Sided
Power:	0.90
Alpha:	0.05
Group Allocation:	Equal (N1 = N2)
Input Type:	Means
$\mu1$:	12.03166667
$\mu2$:	26.55866667
σ :	3.888577546

AVF+AAV-Vector and AVF+AAV-LncDACH1

Alternative Hypothesis: $H_1: \delta = \mu_1 - \mu_2 \neq 0$

Target Power	Actual Power	N1	N2	N	μ_1	μ_2	δ	σ	Alpha
0.90	0.91880	4	4	8	27.9	16.2	11.6	4.1	0.050

Dropout-Inflated Sample Size

Dropout Rate	Sample Size			Dropout-Inflated Enrollment Sample Size			Expected Number of Dropouts		
	N1	N2	N	N1'	N2'	N'	D1	D2	D
10%	4	4	8	5	5	10	1	1	2

Design Tab

Solve For:	Sample Size
Alternative Hypothesis:	Two-Sided
Power:	0.90
Alpha:	0.05
Group Allocation:	Equal (N1 = N2)
Input Type:	Means
μ_1 :	27.88933333
μ_2 :	16.23966667
σ :	4.061284382

AVF+CTRL and AVF+CKO-LncDACH1

Alternative Hypothesis: $H_1: \delta = \mu_1 - \mu_2 \neq 0$

Target Power	Actual Power	N1	N2	N	μ_1	μ_2	δ	σ	Alpha
0.90	0.99843	3	3	6	24.4	52.2	-27.8	4.9	0.050

Dropout-Inflated Sample Size

Dropout Rate	Sample Size			Dropout-Inflated Enrollment Sample Size			Expected Number of Dropouts		
	N1	N2	N	N1'	N2'	N'	D1	D2	D
10%	3	3	6	4	4	8	1	1	2

Design Tab

Solve For:	Sample Size
Alternative Hypothesis:	Two-Sided
Power:	0.90
Alpha:	0.05
Group Allocation:	Equal (N1 = N2)
Input Type:	Means
μ_1 :	24.40533333
μ_2 :	52.15833333
σ :	4.928695602

2. The delivery of the AAV is not sufficiently addressed. It is done by qPCR but needs to be done with in situ localisation for the lncRNA, plus a control reporter gene AAV should be used to show location and duration of expression (GFP or lacZ, for

example). This is essential to show the location of expression of the lncRNA to understand the mechanisms and functional effects.

Response: As per your suggestions, we performed additional experiments using lncRNA fluorescence in situ hybridization techniques in mice transfected with AAV-LncDACH1 and AAV-Vector to verify the transfection efficiency. Moreover, we also examined the expression and subcellular distribution of GFP at different time points after transfection with AAV-GFP in mice as additional supporting data for transfection efficiency under our experimental conditions.

A Fluorescence in situ hybridization to detect the expression level and localization of LncDACH1 in mice transfected with AAV-Vector and AAV-LncDACH1.

B The expression levels and localization of GFP in mice transfected with AAV-GFP as compared to those in control mice were measured by immunofluorescence assay 4 and 8 weeks after transfection.

3. In situ localisation for lncRNADACH1 is required to show endogenous levels during disease progression in vivo and in patient samples.

Response: Thank you for your wonderful suggestions. To address your concern, we performed additional FISH experiments detecting endogenous levels and the changes in expression levels of LncDACH1 over the course of the disease progression in mice and in patient samples.

A The expression levels and localization of LncDACH1 and α -SMA in the veins of AVF mice on days 0 and 14 days were detected by fluorescence in situ hybridization and immunofluorescence assay, respectively.

B FISH and immunofluorescence assays to detect the expression level and localization of LncDACH1 and α -SMA in human preoperative veins and stenosis veins.

4. Nuclear/cytoplasmic expression levels need to be quantified with additional assays and controls for the C/N portions to ensure the techniques are applied correctly. This is also relevant to the siRNA - in that setting does that affect both N and C portions or just Cytoplasmic? this is important to understand the mechanism of action.

Response: Thank you for your constructive suggestions. We measured the relative levels of LncDACH1 in purified VSMC nuclear and cytoplasmic RNA samples using qPCR methods. For this experiment, we selected GAPDH as the internal reference for the cytoplasm and U6 as the internal reference for the nucleus, as well documented by published studies ¹⁻⁴. We found that of the total levels of GAPDH mRNA, the cytoplasmic portion accounted for 83.6% and the nuclear portion for 16.4%. In contrast, the majority of U6 mRNA was found in the nuclear RNA samples (76.9%) with the cytoplasmic U6 mRNA accounting for only 23.1% of the total levels. This result is an indication of the successful separation of the nucleus and cytoplasmic RNAs in our samples. To further ensure this point, we performed additional experiments. We compared the relative levels of Matal1 which is considered an nuclear internal reference in many studies ¹⁻⁴. As anticipated, Matal1 level was found much lower in the cytoplasmic RNA samples (7.8%) than the nuclear ones (92.2%).

In regard to the effects of siRNA on the target mRNAs in the nuclear and cytoplasmic portions, we refer to the relevant literature. Lennox et al. suggested that siRNA can effectively inhibit the expression of LncRNA ⁵, and for LncRNAs primarily localized in the nucleus, the silencing effect of siRNA is less effective and for LncRNAs localized in both cytoplasm and nucleus, siRNA effectively knock down the targets in both locations. In the present study, LncDACH1 was found present in both cytoplasm and nucleus of VSMCs, and was therefore expected to be silenced equally well in both locations. To verifying this notion, we performed qPCR experiments and found that siRNA significantly reduced LncDACH1 levels in the cytoplasmic fraction, as well as in the nuclear fraction, further verifying that siRNA effectively knocked down LncDACH1 in both nucleus and cytoplasm.

A mRNA levels of LncDACH1, GAPDH, U6 and Matal1 in VSMC cytoplasm and nuclear fractions were measured by qPCR, respectively.

B Application of qPCR to detect the silencing efficiency of si-LncDACH1 in the nuclear and cytoplasm fractions of VSMC, respectively.

*p<0.05; **p<0.01; ***p<0.001. ****p<0.0001; ns: non-significant.

5. In the siRNA experiments (mixing), please be clear on the total doses of siRNA in each group.

Response: Thank you for your suggestions. We have provided the correct concentrations of siRNA in the corresponding Method section (please see below for the correction).

Revision of the manuscript

reporter GV238 vector (GeneChem). Transfection of siRNA or pcDNA into VSMCs was performed with Lipo2000 (Invitrogen). LncDACH1-siRNA were transfected into cells for LncDACH1 knockdown at a final concentration of 100 nM. HSP90-siRNA were transfected into cells for HSP90 knockdown at a final concentration of 80 nM. SRPK1-siRNA were transfected into cells for SRPK1 knockdown at a final concentration of 60 nM. KLF9-siRNA were transfected into cells for KLF9 knockdown at a final concentration of 60 nM. Scrambled siRNA were transfected into cells for negative control at the same final concentrations as in the corresponding experimental groups.

6. Data in Figure 1 needs improving:

- Data points needed in bar plots for clarity of data

Response: As per your suggestions, we have added the corresponding data points to the statistical map.

- in situ needed for co-localisation with vSMC markers to co-localise the lncRNA in tissue samples.

Response: Co-localization of VSMC marker genes α -SMA and LncDACH1 was confirmed in mouse and human vein specimens, as shown below.

A The expression levels and localization of LncDACH1 and α -SMA in the veins of AVF mice on days 0 and 14 days were detected by fluorescence in situ hybridization and immunofluorescence assay, respectively.

B Application of fluorescence in situ hybridization and immunofluorescence assay to detect the expression level and localization of LncDACH1 and α -SMA in human preoperative veins and stenosis veins, respectively.

- Fig 1 H is poor - one cell shown only.

Response: Thank you for your attention to this issue. To address your concern, we provided below the images capturing multiple cells.

A Fluorescence in situ hybridization was used to detect the distribution of LncDACH1 in VSMC.

- As already stated IH needs controls. e.g. NEAT1. and so on.

Response: As per your suggestions, We have now added NEAT1 as an internal control for RNA samples isolated from nuclei.

A Fluorescence in situ hybridization was used to detect the distribution of NEAT1 in VSMC.

7. In figure 2,

- the plasmid over expression in Fig 2 a is very modest.

Response: Thank you for your attention to this issue. To address your concern, we repeated the experiments aiming to double ensure sufficient levels of LncDACH1 overexpression upon transfecting the LncDACH1-overexpressing plasmids into cells. To this end, we first employed the LncDACH1-overexpressing plasmid containing GFP tag and observed the presence of fluorescence in the cells 48h after transfection. Subsequently, we verified the transfection efficiency using qPCR methods under our experimental conditions and observed as high as 10.6 fold increase in LncDACH1 level relative to non-transfected cells. Such a magnitude of increase of expression is considered more than modest and more than enough for eliciting significant cellular effects. Furthermore, this amplitude of LncDACH1 overexpression is within the same range as that reported in our previously published studies ⁶⁻⁷.

A Application of immunofluorescence to detect the expression level of GFP after transfection with OE-LncDACH1 in VSMC.

- Scale bars needed in images

Response: We highly appreciate your careful review of our manuscript. We have added the scale bar to the corresponding figures.

- the cell proliferation rate appears to be very low in these experiments, what is this?

Response: Thank you for your attention to this issue. The low proliferation rate of VSMCs observed in our experiments is normal and can be adequately explained. First, it is known that when VSMCs are in a normal physiological state with a differentiated phenotype, they essentially have low levels of proliferative capacity, whereas when VSMCs are in a dedifferentiated state, they acquire enhanced proliferative capacity⁸. In our in vitro experiments, we used highly differentiated VSMCs for the control group (CTRL), which is expected to have low proliferation rates. Indeed, using EdU and CCK-8 assays, we observed that VSMCs of the CTRL group had a low proliferation capacity. Moreover, numerous published studies related to VSMC phenotype switching reported low levels of the proliferative capacity in VSMCs⁹⁻¹⁰, consistent with our finding.

- the data in Fig 2 j looks very marginal. please be very clear on stats and replicates (biological).

Response: Thank you for your constructive comments and suggestions. We conducted additional western blot experiments and incorporated the new data to

Figure 2H, I, J, and K. The addition of new data increased the number of experiments from n=3 to n=5.

8. Figure 3

- In A, what are the endogenous levels of the lncRNA? these graphs are all relative and we do not get a realistic assessment of the endogenous levels and thus, the endogenous levels that can be inhibited.

Response: Thank you for your attention to this issue. qPCR provides relative quantification of the transcripts under examination. We therefore are unable to give you the absolute quantification of LncDACH1 transcripts in molar concentration. However, we can provide you the CT values as an alternative way of describing the quantification, as summarized in Table 1 below. The CT values suggest that endogenous LncDACH1 is in a medium level similar to the majority of lncRNAs.

Table 1 The endogenous levels of the LncDACH1

	si-NC	si-LncDACH1	si-NC	si-LncDACH1	si-NC	si-LncDACH1
	-1	-1	-2	-2	-3	-3
LncDACH1(CT)	25.73	27.78	26.70	29.77	25.23	27.01

- Scale bars needed

Response: Thank you for your careful reading of our manuscript. We have added scale bars to the respective figures.

- Similar additional comments to those in Figure 2 regarding data points and replicates.

Response: As stated above, we have increased the number of western blot experiments from n=3 to n=5 in Figure 3H, I, J, and K.

9. In figure 4p, there are no controls at all. This is therefore a negative association and not a controlled experiment.

Response: Thank you for your attention to this issue. In this figure, anti-IgG serves as a negative control, and INPUT is a positive control. This is a common practice for this type of illustration in numerous published studies¹¹⁻¹³. Additionally, we also used GAPDH as an additional negative control for the data. We believe that these controls are sufficient for our data sets.

6. In Figure 5 M, again the data look marginal for the graphs in the green and yellow bars and is this not the important comparison?

Response: We thank you for your wonderful and important suggestion. To address your concern, we have added a group of P+si-SRPK1 vs. P+si-LncDACH1+si-SRPK1 for comparison (please see below for the new data).

A The ability of VSMCs to migrate was tested by Transwell assay.

* $p < 0.05$; ** $p < 0.01$; *** $p < 0.001$. **** $p < 0.0001$; ns: non-significant.

11. In the in vivo experiments, more data are needed in the main figure to understand the effects being observed.

- The vessel morphometry in both experiments needs full analysis.

Response: We are grateful for your many excellent and constructive suggestions that we believe have helped us greatly improve our work. As per your suggestions, we carried out additional analyses on vascular morphology (Neointimal Area, Lumen Area, Lumen Area/Neointimal Area, Neointimal/Media + Adventitia ratio) to make the data more convincing.

A-E Vascular morphology analysis by Neointimal Area, Lumen Area, Lumen Area / Neointimal Area ratio, and Neointima / (Media + Adventitia) ratio for in vivo. * $p < 0.05$; ** $p < 0.01$, *** $p < 0.001$, **** $p < 0.0001$; ns: non-significant.

- Endothelial coverage, SMC layer markers and so on. Proliferation rates too, especially of the SMC layer.

Response: Thank you for your suggestions. We performed additional TUNEL assay and immunohistochemical experiments including labeling the markers for proliferation (Ki-67), fibroblasts (FSP-1), VSMC synthetic phenotype (Vimentin), VSMC contraction phenotype (α -SMA), endothelial cells (CD31), and macrophages (CD68), as you suggested. (please see below for the new images).

C

D

E

A-E TUNEL staining in vivo was detected by immunofluorescence.

F-M The expression levels and localization of Ki67, CD68, CD31, α -SMA, Vimentin, and FSP-1 were determined by immunohistochemistry in vivo.

- Lack of expression and gain of expression need confirming with in situ as well as the PCR.

Response: Thank you for your suggestions. We have conducted additional FISH experiments to demonstrate the changes of LncDACH1 expression levels after various treatments in mice.

A Application of fluorescence in situ hybridization assay to detect the expression level and localization of LncDACH1 in mice transfected with AAV-Vector and AAV-LncDACH1.

B Application of fluorescence in situ hybridization assay to detect the expression level and localization of LncDACH1 in CTRL and CKO-LncDACH1 mice.

- These data are interesting but incomplete. Finally, the mechanistic effects of the lncRNA are incompletely assessed.

Response: LncDACH1 is a LncRNA localized in both the cytoplasm and the nucleus of VSMCs. The main finding of the present study is that the cytoplasmic LncDACH1, but not the nuclear LncDACH1, plays a critical role in regulating the process of AVF neointimal hyperplasia. Mechanically, we uncovered that LncDACH1 targets the HSP90/SRPK1-AKT signaling pathway to control neointimal hyperplasia. Our

conclusion is based on the results obtained from multiple methodologies including quantitative protein profiles and RNA Pull Down. We believed that our experimental results provided sufficient evidence in support of our conclusion on the mechanism of action of LncDACH1. Nonetheless, we entirely agree with you that the present study did not elucidate the precise role and mechanisms of the nuclear LncDACH1 in AVF neointimal hyperplasia, though this part is in our plan for future studies. We also agree with you that without elucidating the role of the nuclear LncDACH1, our understanding of the function of this lncRNA is incomplete. Yet, we are also aware that rigorous studies in the future are required to achieve a comprehensive understanding of the functionalities of LncDACH1.

- The western blots alone are not convincing and n=1 shown.

Response: Thank you for picking up the mistake. As a matter of fact, we performed at least n=3 independent western blots experiments for each data point; however, we mentioned only one set of the data in the main text. We have now corrected this mistake and added two other sets of data to the figure. (Please see below for your information).

Figure 2G

Figure 3G

Figure 4E

Figure 4F

Figure 4G

Figure 4N

Figure 5F

Figure 5G

Figure 5H

Figure 5N

Figure 8F

Figure 8H

REFERENCES

1. Zhu, W. et al. Low Glucose-Induced Overexpression of HOXC-AS3 Promotes Metabolic Reprogramming of Breast Cancer. *Cancer Res.* 82, 805–818 (2022).
2. Yu, Z. et al. Exosomal LOC85009 inhibits docetaxel resistance in lung adenocarcinoma through regulating ATG5-induced autophagy. *Drug Resist. Updat.* 67, 100915 (2023).
3. Gao, R. et al. Long Noncoding RNA Cardiac Physiological Hypertrophy-Associated Regulator Induces Cardiac Physiological Hypertrophy and Promotes Functional Recovery After Myocardial Ischemia-Reperfusion Injury. *Circulation* 144, 303–317 (2021).
4. Li, S. et al. LncRNA Lnc-APUE is Repressed by HNF4 α and Promotes G1/S Phase Transition and Tumor Growth by Regulating MiR-20b/E2F1 Axis. *Adv. Sci.* 8, 2003094 (2021).

5. Lennox. et al. Cellular localization of long non-coding RNAs affects silencing by RNAi more than by antisense oligonucleotides. *Nucleic Acids Res.* 44, 863–877 (2016).
6. Cai, B. et al. Long Noncoding RNA–DACH1 (Dachshund Homolog 1) Regulates Cardiac Function by Inhibiting SERCA2a (Sarcoplasmic Reticulum Calcium ATPase 2a). *Hypertension* (2019) .
7. Cai, B. et al. Targeting LncDACH1 promotes cardiac repair and regeneration after myocardium infarction. *Cell Death Differ.* 27, 2158–2175 (2020).
8. Frismantiene, A. et al. Smooth muscle cell-driven vascular diseases and molecular mechanisms of VSMC plasticity. *Cell. Signal.* 52, 48–64 (2018).
9. Ye, C. et al. Inhibition of miR-135a-5p attenuates vascular smooth muscle cell proliferation and vascular remodeling in hypertensive rats. *Acta Pharmacol. Sin.* 42, 1798–1807 (2021).
10. Zheng, X. et al. TAK1 accelerates transplant arteriosclerosis in rat aortic allografts by inducing autophagy in vascular smooth muscle cells. *Atherosclerosis* 343, 10–19 (2022).
11. Yang, H. et al. Hypoxia inducible lncRNA-CBSLR modulates ferroptosis through m6A-YTHDF2-dependent modulation of CBS in gastric cancer. *J. Adv. Res.* 37, 91–106 (2022).
12. Li, P. et al. Inhibition of long noncoding RNA HIF1A-AS2 confers protection against atherosclerosis via ATF2 downregulation. *J. Adv. Res.* 26, 123–135 (2020).
13. Shang, Z. et al. LncRNA PCAT1 activates AKT and NF- κ B signaling in castration-resistant prostate cancer by regulating the PHLPP/FKBP51/IKK α complex. *Nucleic Acids Res.* 47, 4211–4225 (2019).

Reviewer #2 (Remarks to the Author):

Overall summary:

This paper studies the role of LncDACH1 in venous neointimal hyperplasia un mouse with AVFs. The author use cell culture experiments to investigate the role of LncDACH1 in multiple different in vitro experiments. The authors place AVFs in mouse and show data for vascular remodeling. I have several questions for the authors with respects to the data that is presented:

mouse AVFs:

1. The authors need to show data for additional measures including neointima area, media + adventitia area, lumen vessel area (LVA), Ki-67, and TUNEL.

Response: Thank you for your great suggestions. We have added additional analysis on blood vessel morphology. We also performed additional TUNEL and immunohistochemical experiments with Ki67 staining (The result images of Ki67 staining are presented in the next response), as you suggested. (Please see below for the new images).

B**C****D**
E**F**

A-E TUNEL staining in vivo was detected by immunofluorescence.

F-J Vascular morphology analysis by Neointimal Area, Lumen Area, Lumen Area / Neointimal Area ratio, and Neointima / (Media + Adventitia) ratio for in vivo.

* $p < 0.05$, ** $p < 0.01$, *** $p < 0.001$, **** $p < 0.0001$; ns: non-significant.

2. Also, they should show the other cellular phenotypes that are present in the vessel wall including fibroblast, SMC phenotypes (contractile vs synthetic), and endothelial cells, and macrophages.

Response: Thank you for your suggestions. We performed additional immunohistochemical experiments including labeling the markers for fibroblasts (FSP-1), VSMC synthetic phenotype (Vimentin), VSMC contraction phenotype (α -SMA), endothelial cells (CD31), and macrophages (CD68), as you suggested. (please see below for the new images).

A-H The expression levels and localization of Ki67, CD68, CD31, α -SMA, Vimentin and FSP-1 were determined by immunohistochemistry *in vivo*.

3. Need n/m+a ratio, LVA, LVA/n

Response: As per your suggestion, we have added additional analysis on blood vessel morphology.

A-E Vascular morphology analysis by Neointimal Area, Lumen Area, Lumen Area / Neointimal Area ratio, and Neointima / (Media + Adventitia) ratio for in vivo. * $p < 0.05$; ** $p < 0.01$, *** $p < 0.001$, **** $p < 0.0001$; ns: non-significant.

4. Cell density

Response: Thank you for your attention to this issue. In this study, when performing cell culture and passaging, we ensured that the cells were passaged at 80-90% of cell growth density to ensure the normal growth state of the cells.

5. **a** The removal of mouse that were only successful biases the data and we need to know what did the other animals staining features with respect to the readouts. How many animals were there? **b** Also, the animals have normal kidney function and this limits the utility of the animal model.

Response a: Thank you for your important suggestions and comments. In our study, we confirmed the successful development of the AVF model by observing alternating arterial and venous flow in the IVC. To address your concern, we have excluded data from the animal models that were not successfully established due to surgical operations on the same day.

In addition, some of the AVF mice died within 24 hours postoperatively due to increased venous return volume or acute thrombosis or other causes. Since we need to

observe the effect of LncDACH1 on AVF NIH in the AVF model and collected the samples on day 14 , we also need to exclude animals that died within 24 hours after surgery to ensure the accuracy and rigor of the animal experiments. We did not perform additional staining experiments on the excluded animal samples. We keep all notes and records about the exclusion and the reasons for the exclusion of experimental animals.

The sample size of animals used in our study is summarized in the table shown below.

Revision of the manuscript

buprenorphine. AVF veins were collected on postoperative Days 0, 7, 14 and 21. Since we need to conduct animal experiments based on the successfully establishment of the AVFs, only successfully establishment AVFs were used in this study, and unsuccessfully establishment AVFs due to surgical technique and death within 24 hours after surgery, etc. were excluded.

Table 1 Sample size of animal model

Experimental group	Initial sample size	A	B	Final sample size
0 day	10	/	/	10
14 days (AVF)	10	1	0	9
AVF + CTRL	10	2	0	8
AVF + CKO-LncDACH1	10	1	1	8
AVF + AAV-Vector	10	0	1	9
AVF + AAV- LncDACH1	10	1	1	8

Exclusion reasons: **A** Surgery operation; **B** Death within 24 hours after surgery

Response b: We do agree with you on the point that the animals with normal renal function may limit the utility of our experiments. Currently, the commonly used method is to remove the kidney 5/6 and detect the blood creatinine and urea nitrogen about 28 days after surgery to determine whether the chronic kidney disease model is successfully established. While this model is widely used for its successful simulation of progressive renal failure in humans, it has a high mortality rate ¹. For our study, we need to perform a secondary operation on the chronic kidney disease model. Since the general state of mice with chronic kidney disease is worse than normal ones, the ability of the mice after surgery to resist stress such as surgical trauma is rather poor, which is a big challenge to the final survival rates of model mice.

In AVF related studies published in recent years, many of the AVF mouse models were established with normal mice, to ensure survival rate following secondary

surgery in mice with chronic kidney disease ²⁻⁶. Before the successful development of methods for solving the problem of lower survival rate following secondary surgery, the use of mice with normal kidney function is a compromised but acceptable alternative.

6. Human AVF Sample Collection:

What are the demographics, age, comorbidities

Response: We have now provided the basic characteristics and demographics as well as other relevant information about the human participants from how the tissue samples used in our study were collected. (Please see Table 2 below)

Table 2 Characteristics of the participants

Patient Characteristics	Vein (n=3)	Stenoisis Vein (n=3)	P Value
Age at time of surgery, years	55.33±4.93	58.67±4.73	NS
Sex	3M;0F	3M;0F	NS
Active smoker (%)	0	0	NS
Hypertension (%)	66.7	66.7	NS
Diabetes (%)	33.3	66.7	NS
Coronary arterial disease (%)	33.3	33.3	NS
Venous thromboembolic disease (%)	0	0	NS

M, male; F, female; NS, no significant ($P > 0.05$).

7. Figures: OK

Response: Thank you for your approval.

References

1. Yin-Wu Bao. et al. Kidney disease models: tools to identify mechanisms and potential therapeutic targets. *Zool Res.* 39, 72–86 (2018).
2. Chang, C.-J. et al. Osteopontin mediation of disturbed flow–induced endothelial mesenchymal transition through CD44 is a novel mechanism of neointimal hyperplasia in arteriovenous fistulae for hemodialysis access. *Kidney Int.* 103, 702–718 (2023).
3. Taniguchi, R. et al. Endothelial Cell TGF- β (Transforming Growth Factor-Beta) Signaling Regulates Venous Adaptive Remodeling to Improve Arteriovenous

- Fistula Patency. *Arterioscler. Thromb. Vasc. Biol.* 42, 868–883 (2022).
4. Somarathna, M. et al. Nitric oxide releasing nanomatrix gel treatment inhibits venous intimal hyperplasia and improves vascular remodeling in a rodent arteriovenous fistula. *Biomaterials* 280, 121254 (2022).
 5. Hashimoto, T. et al. Stimulation of Caveolin-1 Signaling Improves Arteriovenous Fistula Patency. *Arterioscler. Thromb. Vasc. Biol.* 39, 754–764 (2019).
 6. Feng, W. et al. The Transcription Factor E26 Transformation–Specific Sequence-1 Mediates Neointima Formation in Arteriovenous Fistula. *J. Am. Soc. Nephrol.* 25, 475–487 (2014).

Reviewer #3 (Remarks to the Author):

This is a well done study that provides new information regarding the mechanisms of NIH in AVF. This study provides evidence of a potential intervention for NIH in AVF, namely the LncDACH1 pathway, which is significant due to the short-lived nature of AVF.

Response: We highly appreciate your very positive comments on our work and thank you for your constructive suggestions.

1. While the authors clearly indicate that more studies are needed to confirm these findings, the current study draws reasonable conclusions from the data generated.

Response: We thank you very much for your appreciation of our work.

2. As stated in the manuscript the this study is lacking in the number of human samples examined due to the scarcity of such tissue. The authors acknowledge this limitation.

Response: We entirely agree with you that the number of human specimens used in the present study is small, which is one of the limitations of our work. To this end, we are currently making continuous efforts in collecting more samples from multicenter clinical sources to further our investigations on the related subject as well as to compensate for the limitation.

3. The methodology is sound and for the most part the information supplied is sufficient for the work to be replicated. In line 132, the authors at the specific ratio is absent and needs to be stated.

Response: We are grateful for your very positive comments on our study. As per your suggestion, we have now provided the information about the specific mixing ratios in the Method section.

Revision of the manuscript

Generation of Adeno-associated Virus overexpressing LncDACH1 mice. An

adeno-associated virus vector carrying LncDACH1 (OE-LncDACH1) was constructed by

GeneChem (China). The adeno-associated virus was mixed with the biogel Pluronic F-127

(Merck) at 1:5 ratio and then topically applied to the surface of the vessels for 5 min; then, the

gel was allowed to solidify. Four weeks later, the vessel tissues were isolated, and transfection

efficiency was verified by qRT-PCR.

4. The authors also state that animal studies were performed until significance was reached. This design is not standard and a power analysis should have been performed to determine animal number prior to initiation of the study.

Response: Thank you for your constructive and important suggestions. We performed our power analysis separately for in vivo experiments using PASS 15 software, with an assumption of Alpha = 0.05 and Target Power > 0.90. With such a setting, only n=5 experimental animals per group are required for adequate and satisfactory statistical analyses. Yet, to further address your concern and ensure the accuracy of our experimental conclusions, we conducted additional in vivo experiments, adding up to a minimum of n=8 animals per group. The addition of the new data does not change our original conclusions.

0 days and 14 days

Alternative Hypothesis: $H_1: \delta = \mu_1 - \mu_2 \neq 0$

Target Power	Actual Power	N1	N2	N	μ_1	μ_2	δ	σ	Alpha
0.90	0.91994	3	3	6	12.0	26.6	-14.5	3.9	0.050

Dropout-Inflated Sample Size

Dropout Rate	Sample Size			Dropout-Inflated Enrollment Sample Size			Expected Number of Dropouts		
	N1	N2	N	N1'	N2'	N'	D1	D2	D
10%	3	3	6	4	4	8	1	1	2

Design Tab

Solve For:	Sample Size
Alternative Hypothesis:	Two-Sided
Power:	0.90
Alpha:	0.05
Group Allocation:	Equal (N1 = N2)
Input Type:	Means
μ_1 :	12.03166667
μ_2 :	26.55866667
σ :	3.888577546

AVF+AAV-Vector and AVF+AAV-LncDACH1

Alternative Hypothesis: $H1: \delta = \mu_1 - \mu_2 \neq 0$

Target Power	Actual Power	N1	N2	N	μ_1	μ_2	δ	σ	Alpha
0.90	0.91880	4	4	8	27.9	16.2	11.6	4.1	0.050

Dropout-Inflated Sample Size

Dropout Rate	Sample Size			Dropout-Inflated Enrollment Sample Size			Expected Number of Dropouts		
	N1	N2	N	N1'	N2'	N'	D1	D2	D
10%	4	4	8	5	5	10	1	1	2

Design Tab

Solve For:	Sample Size
Alternative Hypothesis:	Two-Sided
Power:	0.90
Alpha:	0.05
Group Allocation:	Equal (N1 = N2)
Input Type:	Means
μ_1 :	27.88933333
μ_2 :	16.23966667
σ :	4.061284382

AVF+CTRL and AVF+CKO-LncDACH1

Alternative Hypothesis: $H1: \delta = \mu_1 - \mu_2 \neq 0$

Target Power	Actual Power	N1	N2	N	μ_1	μ_2	δ	σ	Alpha
0.90	0.99843	3	3	6	24.4	52.2	-27.8	4.9	0.050

Dropout-Inflated Sample Size

Dropout Rate	Sample Size			Dropout-Inflated Enrollment Sample Size			Expected Number of Dropouts		
	N1	N2	N	N1'	N2'	N'	D1	D2	D
10%	3	3	6	4	4	8	1	1	2

Design Tab

Solve For:	Sample Size
Alternative Hypothesis:	Two-Sided
Power:	0.90
Alpha:	0.05
Group Allocation:	Equal (N1 = N2)
Input Type:	Means
μ_1 :	24.40533333
μ_2 :	52.15833333
σ :	4.928695602

5. Another omission is the statement of a clear hypothesis in the introduction. While the authors do refer to their hypothesis later in the text, the hypothesis needs to be stated upfront.

Response: Thank you for your wonderful suggestion. We have now clearly stated our hypothesis in the Introduction section, as you suggested.

Revision of the manuscript

attenuated AVF NIH. Mechanistically, LncDACH1 inhibited serine/arginine-rich splicing factor protein kinase 1 (SRPK1) and heat shock protein 90 (HSP90) to regulate p-AKT. Using RNA Immunoprecipitation (RIP) assay, we found that LncDACH1 directly bound SRPK1 but not HSP90. Our data also uncovered that LncDACH1 regulated nuclear translocation of SRPK1 by targeting HSP90. These results prompted us to posit that LncDACH1 controls the nuclear translocation of SRPK1 by regulating the binding and interactions between HSP90 and SRPK1, which may be a key mechanism underlying the cellular function of LncDACH1. Additionally, the transcription factor KLF9 acted as a transactivator to positively regulate LncDACH1 transcription by binding directly to the promoter region of the LncDACH1 gene.

6. There is also a confusing statement in line 295 stating "LncDACH1 was more significantly downregulated in mouse VSMCs during dedifferentiation". Was this tested statistically or is it simply anecdotal observation of the data?

Response: An excellent question! We provide you with our answers to address your concern from the following two aspects. Firstly, we observed that the greatest downregulation of LncDACH1 in mouse and human VSMCs, relative to their respective controls, was observed 48h following the application of PDGF-BB (10 ng/mL). Secondly, the magnitude of LncDACH1 downregulation in mouse VSMCs was significantly greater than in human VSMCs. Our statement "LncDACH1 was more significantly downregulated in mouse VSMCs during dedifferentiation" was based upon these observations.

A LncDACH1 expression levels in mouse and humans VSMC were measured by q-PCR after stimulation with PDGF-BB (10 ng/mL) for 48h. * $p < 0.05$; ** $p < 0.01$; *** $p < 0.001$. **** $p < 0.0001$; ns: non-significant.

7. Line 612 has a typographical error

Response: Thank you for your careful reading of our manuscript. We have corrected the typographical error you picked up and rearranged the Acknowledgments.

Revision of the manuscript

Acknowledgments

This study was supported by the National Natural Science Foundation of China (82170756) and the Technology Research and Development Program of Heilongjiang (GY2019YF0165).

We thank J.C. of the Animal Center of the Second Affiliated Hospital of Harbin Medical University for technical support for the establishment of the animal model for this study.

REVIEWER COMMENTS

Reviewer #1 (Remarks to the Author):

The authors have made several detailed responses and many of these are acceptable. However, there are remaining issues.

1. The additional repeat western blots are welcome, yet needed careful analysis and the whole blots shown.
2. The AAV overexpression needs co-localisation in both the context for SMC (where it is needed) and expression in other cells of the AVF, particularly fibroblasts and EC/immune cells). It is important that the response is driven by the SMC compartment as indicated by the study of this lncRNA in this cell type.
3. The lncRNA in situ is not complete. There are no control probes, which are essential. And also co-localisation is largely missing.
4. Further, in one image now show of multiple SMC in vitro, the distribution is odd – sometimes nuclear, sometimes at the membrane. Thus, it is not reassuring that the probes are working specifically.
5. Mechanism is still largely incomplete.

It is an interesting finding and speaks to the importance of VSMC lncRNA, but there are remaining issues above to tackle regarding the provenance of the data and the mechanism of action and location.

Reviewer #2 (Remarks to the Author):

Overall summary:

The authors have submitted a revised manuscript. This paper studies the role of lncDACH1 in venous neointimal hyperplasia in mice with AVFs. The authors use cell culture experiments to investigate the role of lncDACH1 in multiple different in vitro experiments. The authors have very strong mechanistic data in cell culture combined with molecular techniques and bioinformatics. The major weakness is in the lack of description of the human data, mice data (choice of mice AVF model) and how the information is provided, lack of chronic kidney disease, the short time point after AVF placement, lack of hemodynamics, lack of confirmation of cell culture data in the mice AVFs, and a weak discussion concerning the murine findings.

The authors place AVFs in mice and show data for vascular remodeling. They have addressed most of my comments, however they have not included any of the new mice data in the paper. Also, they need to include more discussion on the mouse model, the weaknesses, and the fact that the results are only at 14 days. Many mice AVF papers provide longer timepoint-28 days after AVF creation (PMID: 33893223; PMCID: PMC8425661). Authors need to show in the mice experiments that smooth muscle cell switching is occurring using the phenotypic markers such as OPN, vimentin, sm22alpha as they used in cell culture experiments. They have not done this.

The authors show new data for mice morphometry which is not included in the methods, results or discussion.

Abstract: ok

Introduction: OK

Methods:

P6, line 113- is tamoxifen given ip? It says injected

KO: did they confirm cell specific deletion in conditional ko mice

Was there deletion of the gene in endothelial cells

Smc Phenotype conversion in ko AVFs needs to be performed. Specifically showing that synthetic to contractile smcs are decreasing.

P7, line 134 biogel- what percent biogel Pluronic f127 did they use

What does the ratio of 1.5 mean

We need to know how much AAV vector was used

Histology

How many sections per animal were analyzed for morphometry and stains

The authors say that mean neointima hyperplasia thickness was determined as previously described.

There is no reference to this statement

Statistical methods:

Major weakness: The authors need to use a non-parametric statistical test instead of Student T test.

They need to use Bonferroni correction where applicable. The test used should be indicated at the bottom of all figure legends.

Results:

Human AVFs:

The demographics and characteristics of the patients enrolled in the trial needs to be added in the results section. This is missing and I don't see it other than in the rebuttal letter. This belongs in the paper.

Mice AVFs:

The authors need to provide a section on surgical outcomes of the mice avfs including the number of animals used to generate the data. This is missing and it makes it hard to know exactly what was done.

Authors need to show in the mice experiments that smooth muscle cell switching is occurring using the phenotypic markers such as OPN, vimentin, sm22alpha as they used in cell culture experiments . They have not done this.

The authors need to add the additional morphometry data they provided in the rebuttal to reviewer 2 with smooth muscle cell, CD 68, tunnel and proliferation into the results section.

Fig. 1: In the in vitro experiments with human and mouse, SMC's, the authors do not show that the smooth muscle cells did differentiate. They need to do additional experiments, identifying the phenotype of the smooth muscle cells after stimulation with PDGF.

Discussion:

The authors need to expand their discussion to highlight the findings in the murine model and compare this to other therapeutics used in mice AVFs. This is the major strength of the paper and provides potential for translation.

Significant weakness: Add references to other manuscripts using the murine model. This will strengthen

the paper significantly.

Limitations:

The authors need to add the following limitations. One male mice were only used. Add in limitation that female mice were only used but female mice with AVFs have been shown to have distinct different response to AVF placement than male mice. Ref: PMID: 32757791; PMCID: PMC7660821.

Mice with normal kidney function were used instead of those with chronic kidney disease Ref: PMID: 20881937; PMID: 27011423; PMCID: PMC4803510.

Also, the authors used aorta caval fistula model, which is a model for Vascular maturation, where, as the carotid artery to jugular vein model is a model for neointimal hyperplasia.

Lack of ultrasound hemodynamic data.

Figures:

Fig. 1: Fig. 1A: stenosis is spelled wrong in the title. Fig.1A, has a box on the low mag showing the adventitia, whereas the high mag shows the neointimal hyperplasia. This needs to be corrected. The human sections need to be quantified beyond just measuring the thickness of the wall. After AVF placement, all AVFs will thicken which is part of maturation. They should measure and provide the following data:

ratio of neointima area /media+adventitia, Lumen vessel area

Fig. 8: Major weakness: morphometry data needs to be added as stated above. This is included in response to number 11. The TUNEL, proliferation, endothelial data, Cd68, sma, fsp-1, and vimentin needs to be quantified and added to the results and discussion. What markers in mice are being used to show smooth muscle cell switching?

Reviewer #3 (Remarks to the Author):

This reviewer is comfortable with the authors response to the previous reviews.

REVIEWER COMMENTS

Reviewer #1 (Remarks to the Author):

The authors have made several detailed responses and many of these are acceptable. However, there are remaining issues.

1. The additional repeat western blots are welcome, yet needed careful analysis and the whole blots shown.

Response: Thank you for your recognition and suggestions. We have replaced the reanalyzed the western blots data to the manuscript and provided the corresponding source data.

Fig. 2g

Fig. 3g

Fig. 4e

Fig. 4f

Fig. 4g

Fig. 4n

Fig. 5a

Fig. 5d

Fig. 5e

Fig. 5f

Fig. 5g

Fig. 5h

Fig. 5n

Fig. 6b

Fig. 6c

Fig. 6c

Fig. 8g

Fig. 8h

Fig. 8m

Fig. 8n

Fig. S1g

Fig. S11a

Fig. S11b

Fig. S4a

Fig. S6a

Fig. S8a

Fig. S8b

2. The AAV overexpression needs co-localisation in both the context for SMC (where it is needed) and expression in other cells of the AVF, particularly fibroblasts and EC/immune cells). It is important that the response is driven by the SMC compartment as indicated by the study of this lncRNA in this cell type.

Response: We are grateful for your constructive suggestions that we believe have helped us greatly improve our study. Following your suggestion, we performed co-localization analyses of the fibroblast marker FSP-1, the endothelial cell marker CD31 and the macrophage marker CD68 with LncDACH1 in AAV overexpression mice, respectively. We found no significant co-localization of LncDACH1 with CD31, FSP-1 and CD68. As you said, this is crucial to understand the mechanism of LncDACH1 at the cell type.

A Application of fluorescence in situ hybridization and immunofluorescence assay to detect the co-localization of LncDACH1 and CD31 in AAV-LncDACH1 mice.

B Application of fluorescence in situ hybridization and immunofluorescence assay to detect the co-localization of LncDACH1 and FSP-1 in AAV-LncDACH1 mice.

C Application of fluorescence in situ hybridization and immunofluorescence assay to detect the co-localization of LncDACH1 and CD68 in AAV-LncDACH1 mice.

3. The lncRNA in situ is not complete. There are no control probes, which are essential. And also co-localisation is largely missing.

Response: Thank you for your attention to this issue. We have added negative control probes for each group of fluorescence in situ hybridization experiments.

To clarify the co-positioning situation, we have added additional analyses of co-localization of LncDACH1 with α -SMA in mouse and human samples by Image J. The results showed that the co-localization of LncDACH1 with α -SMA was clearer in the mouse 0 days group and the human vein group; compared with the mouse 14 days

group and the human stenosis group, the co-localization was more obvious in the mouse 0 days group and the human vein group.

C

D

E

A-D Detection of staining results of negative control probes in mouse and human samples using fluorescence in situ hybridization experiments.

E Co-localization analysis of LncDACH1 with α -SMA in mouse and human samples by Image J (The more evenly the fluorescence intensity is distributed on both sides of the central axis indicates better co-localization).

4. Further, in one image now show of multiple SMC in vitro, the distribution is odd – sometimes nuclear, sometimes at the membrane. Thus, it is not reassuring that the probes are working specifically.

Response: Thank you for your attention to this issue. To address your concern, first, we reran the in vitro fluorescence in situ hybridization experiments and supplemented them with negative control probes. In the negative control group, we did not see relevant staining results. In the previous positive control experiments we saw NEAT1 expression in the nucleus. The results of the two sets of control experiments were in line with our expectations and demonstrated the correct application of our in situ hybridization experimental system in VSMCs.

This question about the LncDACH1 probe not staining in exactly the same place in the cell has been carefully thought through. We believe that the presence of heterogeneity in cells may be the main cause of this phenomenon. We used the same generation of cells in all in situ hybridization experiments to minimize the problem of heterogeneity caused by different generations of cells. However, a small amount of cellular heterogeneity will inevitably exist even in the same generation of cells. Eventually, a small number of cells may not be localized exactly the same as other cells. For the localization of LncDACH1 in VSMCs, we verified it simultaneously by fluorescence in situ hybridization and RNA nucleoplasmic separation experiments. The results of both methods suggested that LncDACH1 was distributed in both the cytoplasm and nucleus of vascular smooth muscle. Hopefully, our explanation will clear up any concerns you may have had about this issue.

A Fluorescence in situ hybridization was used to detect the distribution of LncDACH1 in VSMC.

5. Mechanism is still largely incomplete.

Response: Thank you for your constructive and important issue. The main finding of the present study is that the cytoplasmic LncDACH1, but not the nuclear LncDACH1, plays a critical role in regulating the process of AVF neointimal hyperplasia. Mechanically, we uncovered that LncDACH1 targets the HSP90/SRPK1-AKT signaling pathway to control neointimal hyperplasia. Our conclusion is based on the results obtained from multiple methodologies including quantitative protein profiles and RNA Pull Down. We believed that our experimental results provided sufficient evidence in support of our conclusion on the mechanism of action of LncDACH1.

However, LncDACH1 is a LncRNA localized in both the cytoplasm and the nucleus of VSMCs. We entirely agree with you that the present study did not elucidate the precise role and mechanisms of the nuclear LncDACH1 in AVF neointimal hyperplasia, though this part is in our plan for future studies. We also agree with you that without elucidating the role of the nuclear LncDACH1, our understanding of the function of this lncRNA is incomplete. We realize that this would be a limitation of this study. Now, we added this part of the limitation to the discussion of the manuscript. Also, we need to conduct the rigorous studies in the future are required to achieve a comprehensive understanding of the functionalities of LncDACH1.

Manuscript Line 658-663

to enhance the integrity of the scientific study. Moreover, LncDACH1 is a LncRNA localized in both the cytoplasm and the nucleus of VSMCs. Although we have uncovered the specific mechanism of action played by cytoplasm LncDACH1 in NIH, we have yet to elucidate the exact role and mechanism of nuclear LncDACH1 in NIH. This may be a limitation to the complete understanding of the LncDACH1 function. We need to conduct rigorous studies in the future are required to achieve a comprehensive understanding of the functionalities of LncDACH1.

It is an interesting finding and speaks to the importance of VSMC lncRNA, but there are remaining issues above to tackle regarding the provenance of the data and the mechanism of action and location.

Reviewer #2 (Remarks to the Author):

Overall summary:

The authors have submitted a revised manuscript. This paper studies the role of LncDACH1 in venous neointimal hyperplasia in mice with AVFs. The authors use cell culture experiments to investigate the role of LncDACH1 in multiple different in vitro experiments. The authors have very strong mechanistic data in cell culture combined with molecular techniques and bioinformatics. The major weakness is in the lack of description of the human data, mice data (choice of mice AVF model) and how the information is provided, lack of chronic kidney disease, the short time point after AVF placement, lack of hemodynamics, lack of confirmation of cell culture data in the mice AVFs, and a weak discussion concerning the murine findings.

The authors place AVFs in mice and show data for vascular remodeling. They have addressed most of my comments, however they have not included any of the new mice data in the paper. Also, they need to include more discussion on the mouse model, the weaknesses, and the fact that the results are only at 14 days. Many mice AVF papers provide longer timepoint-28 days after AVF creation (PMID: 33893223; PMCID: PMC8425661). Authors need to show in the mice experiments that smooth muscle cell switching is occurring using the phenotypic markers such as OPN, vimentin, sm22alpha as they used in cell culture experiments. They have not done this.

The authors show new data for mice morphometry which is not included in the methods, results or discussion.

Abstract: ok

Introduction: OK

Methods:

1. P6, line 113- is tamoxifen given ip? It says injected

Response: Thank you for your careful reading of our manuscript. In fact, tamoxifen is injected into the intraperitoneal of mice. We've fixed this description.

mice were constructed by Biocytogen Co. (China). To obtain smooth muscle cell-specific conditional LncDACH1-knockout mice, LncDACH1 (flox⁺/flox⁺, Cre⁻) mice were hybridization with Cre transgenic mice 80 mg/kg body weight tamoxifen (Sigma) was injected into the intraperitoneal of adult LncDACH1 (flox⁺/flox⁺, Cre⁺) mice once a day, for five days. The control

2. KO: did they confirm cell specific deletion in conditional ko mice. Was there

deletion of the gene in endothelial cells

Response: Thank you for your attention to this issue. To address your concern, we will answer this question from the following two aspects. First, tail genotyping of knockout mice is an important and reliable method to identify whether the target gene has been successfully knocked out. We will perform this method on each batch of mice to ensure that we are using LncDACH1 (floxed/floxed, Cre⁺) conditional knockout mice in subsequent experiments. Next, to further validate the conditional knockout of the gene, we verified this by co-localizing LncDACH1 with the smooth muscle cell marker α -SMA and the endothelial cell marker CD31, respectively. The results showed that in CTRL mice, the co-localization of LncDACH1 with CD31 was not significant; in CKO-LncDACH1 mice, a down-regulation of the expression level of LncDACH1 without substantial changes in co-localization with CD31 could be seen.

C

A Identification of mouse tail genotypes.

B Application of fluorescence in situ hybridization and immunofluorescence for co-localization of LncDACH1 with α -SMA.

C Application of fluorescence in situ hybridization and immunofluorescence for co-localization of LncDACH1 with CD31.

3. SMC Phenotype conversion in ko AVFs needs to be performed. Specifically showing that synthetic to contractile smcs are decreasing.

Response: Thank you for your constructive suggestions. We added additional phenotypic marker validation in mice experiments.

A The expression levels of SM22 α , Opn and Vimentin proteins in vascular tissues of CTRL and CKO-LncDACH1 mouse with established AVF were examined using Western Blot assays.

*, p < 0.05; **, p < 0.01; ***, p < 0.001. ****, p < 0.0001, ns, no significant. A using the Mann-Whitney U test.

4. P7, line 134 biogel- what percent biogel Pluronic f127 did they use, What does the ratio of 1.5 mean, We need to know how much AAV vector was used

Response: Thank you for your attention to this issue. As per your suggestion, we supplemented the amount of AAV used and the ratio to biogel used.

GeneChem (China). Mix adeno-associated virus (10ul; 1.4×10^{12} vg/ml) with biogel PluronicF-127 (50ul) in the proportion and then topically applied to the surface of the vessels for 5 min; then, the gel was allowed to solidify. Four weeks later, the vessel tissues were

Histology

5. How many sections per animal were analyzed for morphometry and stains

Response: Thank you for your attention to this issue. For mice with established AVF, near the point of vessel puncture (approximately 1 mm in length) was sectioned at 5 μ m intervals (approximately 100-120 sections per sample). The section with the greatest neointima area as identified by bright field microscopy and analyzed by HE staining, immunohistochemistry, or immunofluorescence staining. We've added this section to the method.

Histology. Human and mice tissues were first rinsed in PBS and then fixed in 4% PFA. Tissues were embedded in paraffin wax and the section thickness were 5 μ m. For mice with established AVF, near the point of vessel puncture (approximately 1 mm in length) was sectioned at 5 μ m intervals (approximately 100-120 sections per sample). The section with the greatest neointima area as identified by bright field microscopy. The morphology was

6. The authors say that mean neointima hyperplasia thickness was determined as previously described. There is no reference to this statement

Response: Thank you for picking up the mistake. We did not describe it accurately in an earlier manuscript, which we have now corrected.

observed with hematoxylin and eosin (HE) staining. The mean neointimal hyperplasia thickness was measured for each sample by using ImageJ software.

Statistical methods:

7. Major weakness: The authors need to use a non-parametric statistical test instead of Student T test. They need to use Bonferroni correction where applicable. The test used should be indicated at the bottom of all figure legends.

Response: We are grateful for your constructive suggestions that we believe have helped us greatly improve our statistics work. As per your suggestions, we use non-parametric statistical tests instead of the Student's t-test and applied Bonferroni correction for multiple comparisons between groups. In addition, we have added the statistical methods used in the corresponding figure legends for each statistical figure based.

Results:

Human AVFs:

8. The demographics and characteristics of the patients enrolled in the trial needs to be added in the results section. This is missing and I don't see it other than in the rebuttal letter. This belongs in the paper.

Response: Thank you for your attention to this issue. We have added patient demographics and characteristics in the material and methods. And show the complete information in the supplementary material.

Human AVF sample collection. Our studies were approved by the Ethics Committee of the Second Affiliated Hospital of Harbin Medical University. We informed all related patients of the use of these specimens and got the informed consent. Preoperative AVF veins and stenosis AVF veins were collected from different patients. Preoperative AVF veins were collected from uraemic patients who were about to undergo AVF surgery, whereas stenosis AVF veins were collected from veins that were discarded during revision or reconstruction of focal stenosis surgery for AVF treatment. The patient demographics and characteristics are described in Supplementary Table 1. Due to the current availability of endovascular interventions, there are

Supplementary Table 1 . Characteristics of the participants in human samples.

Patient Characteristics	Vein (n=4)	Stenosis Vein (n=4)	P Value
Age at time of surgery, years (median)	57	60	NS
Sex	4M;0F	4M;0F	NS
Active smoker (%)	0	0	NS
Hypertension (%)	75	66.7	NS
Diabetes (%)	50	75	NS
Coronary arterial disease (%)	33.3	50	NS
Venous thromboembolic disease (%)	0	0	NS

M, male; F, female; Mann-Whitney U test. NS, no significant ($P > 0.05$).

Mice AVFs:

9. The authors need to provide a section on surgical outcomes of the mice avfs including the number of animals used to generate the data. This is missing and it makes it hard to know exactly what was done.

Response: Thank you for your constructive comments and suggestions, which is very important for improving the data on surgical mice. We have now provide data on the final sample size for each group of mice that undergo surgery as well as the data that were removed from the final sample size due to surgery operation or death within 24 hours after surgery.

Supplementary Table 2 . Sample size of AVF animal model.

Experimental group	Initial sample size	A	B	Final sample size
0 day	10	/	/	10
14 days (AVF)	10	1	0	9
AVF + CTRL	10	2	0	8
AVF + CKO-LncDACH1	10	1	1	8
AVF + AAV-Vector	10	0	1	9
AVF + AAV- LncDACH1	10	1	1	8

Exclusion reasons: A Surgery operation; B Death within 24 hours after surgery

10. Authors need to show in the mice experiments that smooth muscle cell switching is occurring using the phenotypic markers such as OPN, vimentin, sm22alpha as they used in cell culture experiments . They have not done this.

Response: Thank you for your constructive suggestions. We added additional phenotypic marker validation in mice experiments.

A The expression levels of SM22 α , Opn and Vimentin proteins in vascular tissues of AAV-Vector and AAV-LncDACH1 mouse with established AVF were examined using Western Blot assays.

B The expression levels of SM22 α , Opn and Vimentin proteins in vascular tissues of CTRL and CKO-LncDACH1 mouse with established AVF were examined using Western Blot assays.

*, p<0.05; **, p<0.01; ***, p<0.001. ****, p<0.0001, ns, no significant. **A** and **B** using the Mann-Whitney U test.

11. The authors need to add the additional morphometry data they provided in the rebuttal to reviewer 2 with smooth muscle cell, CD 68, tunnel and proliferation into the results section.

Response: Thank you for your suggestions. We have added the additional morphometry data including smooth muscle cell, CD68, tunnel and proliferation to the corresponding results section. Also, we added the corresponding result images to the manuscript **Fig. 9** (Due to space limitations, the corresponding statistical graphs were added to **Supplementary Fig. 12**). **Manuscript Line 478-508**

12. Fig. 1: In the in vitro experiments with human and mouse, SMC's, the authors do not show that the smooth muscle cells did differentiate. They need to do additional experiments, identifying the phenotype of the smooth muscle cells after stimulation with PDGF.

Response: Thank you for your wonderful suggestions. We validated the phenotypic switching of human smooth muscle cells after PDGF stimulation by additional western blot experiments and added it to the manuscript. For the validation of phenotypic switching in mouse smooth muscle cells, we actually performed the assay in Figure 2g in the CTRL vs P group, however, we neglected to state the results of the phenotypic switching validation here. We have now revised the corresponding results section, and thank you very much for the reminder.

upregulation in this process. First, we determined the successful dedifferentiation of VMSCs after stimulation with PDGF-BB by validating differentiation phenotypic markers α -SMA and SM22 α and de-differentiation phenotypic markers Opn and vimentin. Next, the protein levels of the differentiation markers were upregulated following LncDACH1 overexpression, whereas the protein levels of the dedifferentiation markers were downregulated (Fig. 2g-k). In

A Effect of PDGF-BB stimulation of human VSMCs on the differentiation phenotype marker and dedifferentiation phenotype marker protein levels.

*, p<0.05; **, p<0.01; ***, p<0.001. ****, p<0.0001, ns, no significant. A using the Mann-Whitney U test.

Discussion:

13. The authors need to expand their discussion to highlight the findings in the murine model and compare this to other therapeutics used in mice AVFs. This is the major strength of the paper and provides potential for translation. Significant weakness: Add references to other manuscripts using the murine model. This will strengthen the paper significantly.

Response: Thank you for your constructive suggestions. Based on your suggestions, we have expanded the discussion related to the results of the in vivo experiments. We have also highlighted in the discussion the characteristics of the therapeutic perspectives in the present study compared to other studies (**Manuscript Line 611-643**). Also, we have added the manuscript of the same mouse model as the present study at the appropriate place to enhance the importance of the article.

Next, we established an AVF mouse model based on methods described in previous studies and validated it by H&E staining (Fig. 1d) and morphometric analysis (Fig. 1e; Supplementary Fig. 1c-f)^{14,15,16}. We then performed qRT-PCR and find that LncDACH1

14. Chang, C.-J. *et al.* Osteopontin mediation of disturbed flow-induced endothelial mesenchymal transition through CD44 is a novel mechanism of neointimal hyperplasia in arteriovenous fistulae for hemodialysis access. *Kidney Int.* **103**, 702–718 (2023).
15. Taniguchi, R. *et al.* Endothelial Cell TGF- β (Transforming Growth Factor-Beta) Signaling Regulates Venous Adaptive Remodeling to Improve Arteriovenous Fistula Patency. *Arterioscler. Thromb. Vasc. Biol.* **42**, 868–883 (2022).
16. Hashimoto, T. *et al.* Stimulation of Caveolin-1 Signaling Improves Arteriovenous Fistula Patency. *Arterioscler. Thromb. Vasc. Biol.* **39**, 754–764 (2019).

Limitations:

14. The authors need to add the following limitations. One male mice were only used. Add in limitation that female mice were only used but female mice with AVFs have been shown to have distinct different response to AVF placement than male mice. Ref: PMID: 32757791; PMCID: PMC7660821.

Response: Thank you for your attention to this issue. We have carefully read the literature you have provided. Many current studies suggest that sex differences are an independent risk factor in AVF maturation for reasons that may be related to estrogen levels. In the AVF-related manuscripts involving sex differences, we observed that the main research purpose was to investigate the differential genes and the possible mechanisms of action played by sex differences in the maturation process of AVF¹⁻³. In contrast, for AVF-related manuscripts in which the primary research objective was non-sex-differentiated, most of the literature used male mice as the primary research subjects⁴⁻⁶.

In this paper, our primary research aim is to explore the possible role and mechanism of action of a novel LncRNA-LncDACH1 in AVF maturation. In order to ensure the rigor of our findings, we have standardized the sex of the animals in this study and used male mice to exclude the influence of gender differences in this study. Your highly forward-looking comments also makes us realize that LncDACH1 may play different roles in the maturation process of AVF mice of different sexes. This also points out a clear direction for our future research on LncDACH1 in AVF.

15. Mice with normal kidney function were used instead of those with chronic kidney disease Ref: PMID: 20881937; PMID: 27011423; PMCID: PMC4803510.

Response: We do agree with you on the point that the animals with normal renal function may limit the utility of our experiments. The most important difference

between mice with normal kidney function and chronic kidney disease is whether the blood vessels are exposed to uremic toxins. Recent studies have shown that expression levels of cytokines associated with neointimal hyperplasia formation, including IL-6, TGF- β 1, and TNF- α , are elevated in the uremic environment⁷. In addition, studies have identified increased abundance of markers of oxidative damage (8-hydroxy-20-deoxyguanosine) and lipid peroxidation (4-hydroxy-2-nonenal) in the uremic environment. These factors have been shown to increase levels of potent mitogens, including platelet-derived growth factor (PDGF), endothelin-1, and TGF- β 1, ultimately leading to further neointimal hyperplasia formation⁸.

Our study concludes that LncDACH1 regulates neointimal hyperplasia formation through pro-proliferative and migratory effects on VSMCs in mice with normal kidney function. One of the mechanisms by which uremic toxins would further exacerbate neointimal hyperplasia formation is the elevated expression levels of potent mitogens such as PDGF. Therefore, LncDACH1 may further exert its regulatory role on neointimal hyperplasia formation in the presence of this enhancement mechanism mediated by uremic toxins. At present, we still need to verify this hypothesis by constructing the mice with chronic kidney disease model, which is also a limitation of this experiment. This issue you raised is crucial and constructive, and we have added this part of the limitation discussion to the manuscript. In future experiments, we will follow your suggestions to improve the construction of animal models further to enhance the integrity of the scientific study.

confirm this hypothesis. Another is that animals with normal renal function may limit the utility of our experiments. Our study concludes that LncDACH1 regulates NIH formation through pro-proliferative and migratory effects on VSMCs in mice with normal kidney function. One of the mechanisms by which uremic toxins would further exacerbate NIH formation is the elevated expression levels of potent mitogens such as PDGF. Therefore, LncDACH1 may further exert its regulatory role on NIH formation in this enhancement mechanism mediated by uremic toxins. In future experiments, we will further improve the construction of animal models to enhance the integrity of the scientific study. Moreover, LncDACH1 is a LncRNA localized in

16. Also, the authors used aorta caval fistula model, which is a model for vascular maturation, where, as the carotid artery to jugular vein model is a model for neointimal hyperplasia.

Response: Thank you for your attention to this issue. As you say, although both the

aorta caval fistula model and the carotid artery to jugular vein model have been used to construct AVF models, there are still differences between them. In the carotid artery to jugular vein model, we performed vascular anastomosis of the arteriovenous, and the high pressure arterial blood flowed into the vein and induced venous arterialization (enlargement of the lumen and thickening of the wall). For the aorta caval fistula model, we form a fistula between closely connected arteries and veins by means of a puncture, so that arterial blood can flow through the fistula into the vein and induce venous arterialization.

Differences in the way AVFs are constructed may also lead to differences in the mechanisms by which they form neointima hyperplasia. Venous ECs (endothelial cells) are thought to be the initial activators of AVF neointima hyperplasia formation. In the carotid artery to jugular vein model, ECs near the venous anastomosis were directly impacted by arterial blood flow and experience high, non-physiological blood flow gradients with flow instability. As for the aorta caval fistula model, due to the anatomical structure, the venous vascular endothelium is subjected to a non-direct impact of arterial blood flow compared to the former. This also leads to the different WSS (wall shear stress) to which the venous ECs is subjected in the two models. It has been demonstrated that ECs exert different molecular mechanisms in the process of vascular remodeling when subjected to different WSS⁹⁻¹¹. This suggests that although both AVF models can cause neointima hyperplasia, the specific molecular mechanisms involved may be slightly different. Currently, there are fewer studies in this area, and in the future, we will also include the comparison of the two models in our study to further improve the experiments.

In addition, recently the construction of similar to the aorta caval fistula model has begun to be used in practical clinical applications as an innovative surgical technique. This also suggests that the model has some promise for clinical applications.¹²⁻¹⁴

17.Lack of ultrasound hemodynamic data.

Response: Thank you for your attention to this issue. As you point out, observation of AVF hemodynamics using ultrasound is a reliable means of determining AVF maturation. In general, observation of vascular morphologic features by HE staining is the primary way to clarify AVF maturation. In some studies, ultrasound hemodynamics has also provided additional data support for AVF maturation. In the present study, we were unable to include ultrasound hemodynamic data mainly because the accuracy of the available ultrasound probes in our laboratory was not able to clearly demonstrate the vasculature of the mice as well as the blood flow rate and other data. We are trying to find laboratories with relevant equipment to collaborate with. In the future, we will incorporate ultrasound hemodynamic data into our experiments.

Figures:

Fig. 1:

18.Fig. 1A: stenosis is spelled wrong in the title.

Response: Thank you for picking up the mistake. We've now fixed the spelling.

19.Fig.1A, has a box on the low mag showing the adventitia, whereas the high mag shows the neointimal hyperplasia. This needs to be corrected.

Response: Thank you for your careful reading of our manuscript. We have now corrected this problem. We harmonized the low and high magnification fields of view in the vein groups.

20.The human sections need to be quantified beyond just measuring the thickness of the wall. After AVF placement, all AVFs will thicken which is part of maturation. They should measure and provide the following data:ratio of neointima area /media+adventitia, Lumen vessel area

Response: Thank you for your suggestions. We carried out additional analyses on human vascular morphology (ratio of neointimal/media + adventitia, Lumen vessel area) to make the data more comprehensive (Supplementary Fig. 1).

A Vascular morphology analysis by Neointima / (Media + Adventitia) ratio and Lumen Area / Neointimal Area ratio for in human sample.

* $p < 0.05$, ** $p < 0.01$, *** $p < 0.001$, **** $p < 0.0001$; ns: non-significant. **A** using the Mann-Whitney U test.

Fig. 8:

21. Major weakness: morphometry data needs to be added as stated above. This is included in response to number 11. The TUNEL, proliferation, endothelial data, Cd68, sma, fsp-1, and vimentin needs to be quantified and added to the results and discussion.

Response: Thank you for your constructive suggestions that we believe it is important for a deeper understanding of mice data. We analyzed morphometry data including TUNEL, proliferation, endothelial, CD68, α -SMA, Fsp-1, and Vimentin accordingly and added them to the results (Question 11) and discussion (Question 13).

22. What markers in mice are. Being used to show smooth muscle cell switching?

Response: Thank you for your attention to this issue. This issue was also mentioned in our added discussion. In the in vitro experiments, we selected α -SMA and SM22 α as VSMC differentiation phenotype markers; OPN and Vimentin as VSMC dedifferentiation phenotype markers. A point worth noting here is that in AVF, α -SMA is not only a differentiation phenotype marker for VSMC, but it is also one of the phenotype markers for myofibroblasts. During NIH formation in AVF, fibroblasts distributed in the vascular adventitia are converted into myofibroblasts and thus promote the formation of NIH. This suggests that in AVF, changes in α -SMA expression levels do not directly reflect the differentiation level of VSMC. Therefore, in the in vivo experiments of this study, we selected SM22 α as a VSMC differentiation phenotype marker; OPN and Vimentin were selected as VSMC dedifferentiation phenotype markers.

REFERENCES

1. Cai, C. *et al.* Differences in Transforming Growth Factor- β 1/BMP7 Signaling and Venous Fibrosis Contribute to Female Sex Differences in Arteriovenous Fistulas. *J. Am. Heart Assoc.* **9**, e017420 (2020).
2. Satam, K. *et al.* Sex hormones impact early maturation and immune response in the arteriovenous fistula mouse model. *Am. J. Physiol.-Heart Circ. Physiol.* **325**, H77–H88 (2023).
3. Cai, C. *et al.* Effect of sex differences in treatment response to angioplasty in a murine arteriovenous fistula model. *Am. J. Physiol.-Ren. Physiol.* **318**, F565–F575 (2020).
4. Misra, S. *et al.* Anti Human CX3CR1 VHH Molecule Attenuates Venous Neointimal Hyperplasia of Arteriovenous Fistula in Mouse Model. *J. Am. Soc. Nephrol.* **32**, 1630–1648 (2021).
5. Janardhanan, R. *et al.* Simvastatin reduces venous stenosis formation in a murine hemodialysis vascular access model. *Kidney Int.* **84**, 338–352 (2013).
6. Guo, Q. *et al.* Temporal regulation of notch activation improves arteriovenous fistula maturation. *J. Transl. Med.* **20**, 543 (2022).
7. Wasse, H. *et al.* Inflammation, Oxidation and Venous Neointimal Hyperplasia Precede Vascular Injury from AVF Creation in CKD Patients. *J. Vasc. Access* **13**, 168–174 (2012).
8. Weiss MF, Scivittaro V, Anderson JM. Oxidative stress and increased expression of growth factors in lesions of failed hemodialysis access. *Am J Kidney Dis.* 2001;37:970–980.
9. Franzoni, M. *et al.* Endothelial cell activation by hemodynamic shear stress derived from arteriovenous fistula for hemodialysis access. *Am. J. Physiol.-Heart Circ. Physiol.* **310**, H49–H59 (2016).

10. Chiu, J.-J. & Chien, S. Effects of Disturbed Flow on Vascular Endothelium: Pathophysiological Basis and Clinical Perspectives. *Physiol. Rev.* **91**, 327–387 (2011).
11. Rashad, S. *et al.* Epigenetic response of endothelial cells to different wall shear stress magnitudes: A report of new mechano-miRNAs. *J. Cell. Physiol.* **235**, 7827–7839 (2020).
12. Inston, N., Khawaja, A., Tullett, K. & Jones, R. WavelinQ created arteriovenous fistulas versus surgical radiocephalic arteriovenous fistulas? A single-centre observational study. *J. Vasc. Access* **21**, 646–651 (2020).
13. Lok, C. E. *et al.* Endovascular Proximal Forearm Arteriovenous Fistula for Hemodialysis Access: Results of the Prospective, Multicenter Novel Endovascular Access Trial (NEAT). *Am. J. Kidney Dis.* **70**, 486–497 (2017).
14. Rognoni, C., Tozzi, M. & Tarricone, R. Endovascular versus surgical creation of arteriovenous fistula in hemodialysis patients: Cost-effectiveness and budget impact analyses. *J. Vasc. Access* **22**, 48–57 (2021).

Reviewer #3 (Remarks to the Author):

This reviewer is comfortable with the authors response to the previous reviews.

Response: We highly appreciate to you for carefully reviewing our paper and for your numerous valuable comments!

REVIEWER COMMENTS

Reviewer #1 (Remarks to the Author):

The authors have responded again. I have remaining concerns.

1. Western blots. I asked for whole blots to be shown, but this was not done. I find the response underwhelming, but this is a decision for the journal and their policy on WBs. And some look strange, like 5e.
2. Co-localisation data. Again underwhelming. They show negative staining but no content to the fistula anatomy. The EC one in particular is odd. What are the + cells at the surface? The fibroblast and CD68 staining is poor. Detailed analysis and quantification is needed.
3. Mechanism remains quite relatively the same. While some experiments point to a mechanism, the detailing of that mechanism remains quite superficial.

Reviewer #2 (Remarks to the Author):

The authors have done a good job. They have not included this in the limitations of the paper. This is very important to address. In the response to reviewers they do acknowledge it but do not in their final manuscript.

Limitations:

14. The authors need to add the following limitations. One male mice were only used. Add in limitation that female mice were only used but female mice with AVFs have been shown to have distinct different response to AVF placement than male mice. Ref: PMID: 32757791; PMCID: PMC7660821.

Response: Thank you for your attention to this issue. We have carefully read the literature you have provided. Many current studies suggest that sex differences are an independent risk factor in AVF maturation for reasons that may be related to estrogen levels. In the AVF-related manuscripts involving sex differences, we observed that the main research purpose was to investigate the differential genes and the possible mechanisms of action played by sex differences in the maturation process of AVF1-3. In contrast, for AVF-related manuscripts in which the primary research objective was non-sex-differentiated, most of the literature used male mice as the primary research subjects4-6.

In this paper, our primary research aim is to explore the possible role and mechanism of action of a novel lncRNA-LncDACH1 in AVF maturation. In order to ensure the rigor of our findings, we have standardized the sex of the animals in this study and used male mice to exclude the influence of gender differences in this study. Your highly forward-looking comments also makes us realize that LncDACH1 may play different roles in the maturation process of AVF mice of different sexes. This also points out a clear direction for our future research on LncDACH1 in AVF.

REVIEWER COMMENTS

Reviewer #1 (Remarks to the Author):

The authors have responded again. I have remaining concerns.

1. Western blots. I asked for whole blots to be shown, but this was not done. I find the response underwhelming, but this is a decision for the journal and their policy on WBs. And some look strange, like 5e.

Response: Thank you for your attention to this issue. We truly believe that the concern you raised is of paramount importance. We note your concern about fig 5e. In fact, we performed RNA Pull Down experiments by constructing the full-length fragment A (0-2085bp) of LncDACH1, the fragment E (774-1251bp) of LncDACH1, and the mutated fragment F (774-1251bp mut) of fragment E, respectively, and the pulled down protein products were validated by western blot experiments to identify the true fragments bound by LncDACH1 and SRPK1. To address your concerns, we have re-performed the experiment and provided the corresponding raw data. And we replace this part in manuscript. The conclusions we reached are consistent with the previous ones. In addition, we did our best to re-perform the western blot experiments within the time frame of this limited revision. We followed your suggestion to re-supply the raw data from the whole blot in Figure 3 and show it. The results we derived are consistent with the previous ones. And, we've replaced this result in the manuscript.

In this study, we applied a large number of western blot experiments to verify the changes in the expression of multiple proteins, which consume large amounts of protein. Especially in *in vivo* experiments, since the vascular tissue of each mouse is very small compared to that of parenchymal organs, each mouse can provide only a limited number of protein samples. Our previous approach is to validate numerous proteins simultaneously in a single whole blot to save valuable animal resources, which is also in line with animal ethics to minimize unnecessary waste of animal resources. In addition, the raw data we previous provided were uncropped and unprocessed images after exposure, which is also basically in line with the journal's requirement for the presentation of uncropped versions of any gels or blots.

A RNA Pull Down and Western Blot assays were used to determine the binding of LncDACH1 fragment F to SRPK1.

B Effect of silencing LncDACH1 on VSMC differentiation phenotype markers and dedifferentiation phenotype marker protein levels by Western Blot assay.

2. Co-localisation data. Again underwhelming. ① They show negative staining but no content to the fistula anatomy. ② The EC one in particular is odd. What are the + cells at the surface? ③ The fibroblast and CD68 staining is poor. Detailed analysis and quantification is needed.

Response: Thank you for your constructive comments and suggestions. ① As per your suggestions, we have changed the way of data presentation in the supplemental negative control experiment involved fistula anatomy for the 14-days group. ② Normal vascular wall can be divided into three layers, of which the intimal layer is mainly covered by endothelial cells. We found that CD31, a phenotypic marker for endothelial cells, is primarily expressed in the innermost layer of the vascular wall. In

response to your question and to ensure the accuracy of our data, we performed additional CD31 immunofluorescence staining and found that the expression patterns of CD31 were similar to the documented data in the literature, most abundant in the innermost layer of the vascular wall 1-2. ③ We have addressed the issue concerning the quality of FSP-1 and CD68 staining in the previous round of revision, by showing their co-localization in mice treated with AAV-LncDAHC1 without established AVF. In general, the expression of FSP-1 and CD68 in the vascular tissue of control mice was inherently low, whereas it increased in the counterparts with AVF. We subsequently performed staining and co-localization analyses in the AVF+CKO-LncDAHC1 group of mice and observed that the expression levels of FSP-1 and CD68 were significantly increased in mice with AVF. Moreover, the two phenotypic markers FSP-1 and CD68 did not co-localize with LncDACH1, which is consistent with the data presented in the previous version of our manuscript. We are based on editorial advice to change the color (as substitute blue with cyan and red with magenta) to enhance the clarity of the figures.

A

B**C**
D

E

A Fluorescence in situ hybridization (FISH) display the lack of staining with the negative control probe in mouse samples.

B Immunofluorescence showing the expression and subcellular distribution of CD31 in CTRL mice.

C FISH and immunofluorescence staining showing the overall lack of co-localization

between LncDACH1 and FSP-1 in mice.

D FISH and immunofluorescence staining showing the overall lack of co-localization between LncDACH1 and CD68 in mice.

E Co-localization analysis between LncDACH1 and α -SMA in mouse samples by Image J.

3. Mechanism remains quite relatively the same. While some experiments point to a mechanism, the detailing of that mechanism remains quite superficial.

Response: Thank you for your constructive and important issue. Mechanically, we uncovered that LncDACH1 targets the HSP90/SRPK1-AKT signaling pathway to control neointimal hyperplasia. Our conclusion is based on the results obtained from multiple methodologies including quantitative protein profiles and RNA Pull Down. We believed that our experimental results provided sufficient evidence in support of our conclusion on the mechanism of action of LncDACH1. Nonetheless, we entirely agree with you that the present study did not completely elucidate the precise role and mechanisms of the LncDACH1 in AVF neointimal hyperplasia. We have explored the mechanism in greater depth.

The serine/arginine-rich (SR) proteins are essential splicing regulators required for constitutive pre-mRNA splicing and alternative splicing, while changes in the expression of SR proteins are potentially involved in splicing misregulation in various diseases³. Serine/arginine splicing factor 1 (SRSF1, also named ASF/SF2) is the prototypical member of the highly conserved SR protein family and has been shown to regulate cell proliferation in a variety of diseases and invasion and migration phenotypes in a variety of diseases. In particular, Xie et al. demonstrated that SRSF1 promotes neointimal hyperplasia by facilitating VSMC proliferation and migration after vascular injury⁴⁻⁵. Our group demonstrated in previous studies that LncDACH1 regulates lung fibroblast proliferation and migration by directly binding to SRSF1 and modulating changes in its expression⁶. SRPK1, an upstream regulator of SRSF1, has been shown to bind to SRSF1 and promote its phosphorylation and nuclear translocation, thereby facilitating the variable shearing of SRSF1 in the nucleus⁷. Thus, we speculate that, in addition to regulating VSMC proliferation and migration in the cytoplasm through the AKT signaling pathway, which was demonstrated in this study, SRPK1 may also play a role in the nucleus through the SRPK1-SRSF1 variable shear pathway. On the one hand, after being regulated by LncDACH1, SRPK1 increases its expression level, and binding to SRSF1 prompts the latter to enter the nucleus to exert an alternative splicing mechanism; on the other hand, SRSF1 upregulates its expression after directly binding to LncDACH1, and additional SRSF1 bound to SRPK1 with the same increased expression level further prompts SRSF1 to enter the nucleus to function.

We also speculated that LncDACH1 could form a two-by-two RNA-double protein complex with SRPK1 and SRSF1. Our group found that the lengths of the RNA-binding fragments of LncDACH1 to SRPK1 and SRSF1 were 774-1251 bp and 853-1251 bp, respectively. The highly conserved fragments or structural domains

across species are considered to be the most likely locations at which critical biological functions are performed⁸. Our previous studies confirmed that the sequence of this region of the LncDACH1 fragment is highly conserved between two species, humans and mice, which partly explains the similarity of the sites at which the RNA fragments of SRPK1 and SRSF1 bind to LncDACH1. Therefore, we hypothesized that the simultaneous binding of the highly conserved fragment of LncDACH1 to both SRPK1 and SRSF1 might result in competitive inhibition of LncDACH1 binding by the latter two proteins. The result of this competitive inhibition would prevent some SRPK1 or SRSF1 from binding to LncDACH1, which would also create the possibility of more binding between SRPK1 and SRSF1, and the result would still be that SRPK1 further promotes the nuclear translocation of SRSF1 and exerts pro-proliferative and migratory effects. This hypothesis is based on our team's previous, ongoing research on the mechanism of LncDACH1 and protein binding, and we will fully confirm this hypothesis with additional complete experiments in the future.

REFERENCES

1. Chang, C.-J. et al. Osteopontin mediation of disturbed flow–induced endothelial mesenchymal transition through CD44 is a novel mechanism of neointimal hyperplasia in arteriovenous fistulae for hemodialysis access. *Kidney Int.* 103, 702–718 (2023).
2. Wang, Y. et al. Blocking Notch in Endothelial Cells Prevents Arteriovenous Fistula Failure Despite CKD. *J. Am. Soc. Nephrol.* 25, 773–783 (2014).
3. Long, J. C. & Cáceres, J. F. The SR protein family of splicing factors: master regulators of gene expression. *Biochem. J.* 417, 15–27 (2009).
4. Das, S. & Krainer, A. R. Emerging functions of SRSF1, splicing factor and oncoprotein, in RNA metabolism and cancer. *Mol. Cancer Res.* 12, 1195–1204 (2014).
5. Xie, N. et al. SRSF1 promotes vascular smooth muscle cell proliferation through a $\Delta 133p53/EGR1/KLF5$ pathway. *Nat. Commun.* 8, 16016 (2017).
6. Sun, J. et al. LncRNA DACH1 protects against pulmonary fibrosis by binding to SRSF1 to suppress CTNNB1 accumulation. *Acta Pharm. Sin. B* 12, 3602–3617 (2022).
7. Ghosh, G. & Adams, J. A. Phosphorylation mechanism and structure of serine-arginine protein kinases: SRPK structure and mechanism. *FEBS J.* 278, 587–597 (2011).
8. Ulitsky, I. Evolution to the rescue: using comparative genomics to understand long non-coding RNAs. *Nat. Rev. Genet.* 17, 601–614 (2016).

Reviewer #2 (Remarks to the Author):

The authors have done a good job. They have not included this in the limitations of the paper. This is very important to address. In the response to reviewers they do acknowledge it but do not in their final manuscript.

Limitations:

14. The authors need to add the following limitations. One male mice were only used. Add in limitation that female mice were only used but female mice with AVFs have been shown to have distinct different response to AVF placement than male mice. Ref: PMID: 32757791; PMCID: PMC7660821.

Response: Thank you for your attention to this issue. We have carefully read the literature you have provided. Many current studies suggest that sex differences are an independent risk factor in AVF maturation for reasons that may be related to estrogen levels. In the AVF-related manuscripts involving sex differences, we observed that the main research purpose was to investigate the differential genes and the possible mechanisms of action played by sex differences in the maturation process of AVF1-3. In contrast, for AVF-related manuscripts in which the primary research objective was non-sex-differentiated, most of the literature used male mice as the primary research subjects4-6.

In this paper, our primary research aim is to explore the possible role and mechanism of action of a novel LncRNA-LncDACH1 in AVF maturation. In order to ensure the rigor of our findings, we have standardized the sex of the animals in this study and used male mice to exclude the influence of gender differences in this study. Your highly forward-looking comments also makes us realize that LncDACH1 may play different roles in the maturation process of AVF mice of different sexes. This also points out a clear direction for our future research on LncDACH1 in AVF.

Response: We highly appreciate you for carefully reviewing our paper and for your invaluable comments. We have added a paragraph on the limitation of using only male mice in this study to the discussion of the manuscript. We believe this suggestion is important for improving the quality of the article.

Another limitation is that using only male mice is one of the limitations of this study. Previous studies have shown that sex differences are an independent risk factor for AVF maturation, the reason for which may be related to estrogen levels^{35, 36, 37}. The main objective of this study was to investigate the underlying mechanism of action of the novel LncRNA LncDACH1 in AVF maturation. To ensure the rigor of the

experimental results, we standardized the sex of the animals in this study. We used only male mice to avoid the potential influence of sex differences. However, our study does in no way exclude the possibility that LncDACH1 may play different roles in AVF maturation in mice of different sexes. This issue merits future studies for clarification.

REVIEWERS' COMMENTS

Reviewer #1 (Remarks to the Author):

The authors have done a good job in finalising the remaining issues.